# DATA-EFFICIENT AND INTERPRETABLE TABULAR ANOMALY DETECTION

## ABSTRACT

Anomaly detection (AD) plays an important role in numerous applications. In this paper, we focus on two understudied aspects of AD that are critical for integration into real-world applications. First, most AD methods cannot incorporate labeled data that are often available in practice in small quantities and can be crucial to achieve high accuracy. Second, most AD methods are not interpretable, a bottleneck that prevents stakeholders from understanding the reason behind the anomalies. In this paper, we propose a novel AD framework, DIAD, that adapts a white-box model class, Generalized Additive Models, to detect anomalies using a partial identification objective which naturally handles noisy or heterogeneous features. DIAD can incorporate a small amount of labeled data to further boost AD performances in semi-supervised settings. We demonstrate the superiority of DIAD compared to previous work in both unsupervised and semi-supervised settings on multiple datasets. We also present explainability capabilities of DIAD, on its rationale behind predicting certain samples as anomalies.

## 1 INTRODUCTION

Anomaly detection (AD) has numerous real-world applications, especially for tabular data, including detection of fraudulent transactions, intrusions related to cybersecurity, and adverse outcomes in healthcare. When the real-world tabular AD applications are considered, there are various challenges constituting a fundamental bottleneck for penetration of fully-automated machine learning solutions:

- **Noisy and irrelevant features**: Tabular data often contain noisy or irrelevant features caused by measurement noise, outlier features and inconsistent units. Even a change in a small subset of features may trigger anomaly identification.
- **Heterogeneous features**: Unlike image or text, tabular data features can have values with significantly different types (numerical, boolean, categorical, and ordinal), ranges and distributions.
- **Small labeled data**: In many applications, often a small portion of the labeled data is available. AD accuracy can be significantly boosted with the information from these labeled samples as they may contain crucial information on representative anomalies and help ignoring irrelevant ones.
- **Interpretability**: Without interpretable outputs, humans cannot understand the rationale behind anomaly predictions, that would enable more trust and actions to improve the model performance. Verification of model accuracy is particularly challenging for high dimensional tabular data, as they are not easy to visualize for humans. An interpretable AD model should be able to identify important features used to predict anomalies. Conventional explainability methods like SHAP (Lundberg & Lee, 2017) and LIME (Ribeiro et al., 2016) are proposed for supervised learning and not straightforward to generalize to unsupervised or semi-supervised AD.

Conventional AD methods fail to address the above – their performance often deteriorates with noisy features (Sec. 6), they cannot incorporate labeled data, and cannot provide interpretability.

In this paper, we aim to address these challenges by proposing a **D**ata-efficient **I**nterpretable **AD** framework, **DIAD**. DIAD's model architecture is inspired by Generalized Additive Models (GAMs) and GA$^2$M (see Sec. 3), that have been shown to obtain high accuracy and interpretability for tabular data (Caruana et al., 2015; Chang et al., 2021b; Liu et al., 2021), and have been used in applications like finding outlier patterns and auditing fairness (Tan et al., 2018). We propose to employ intuitive notions of Partial Identification (PID) as an AD objective and learn them with a differentiable GA$^2$M

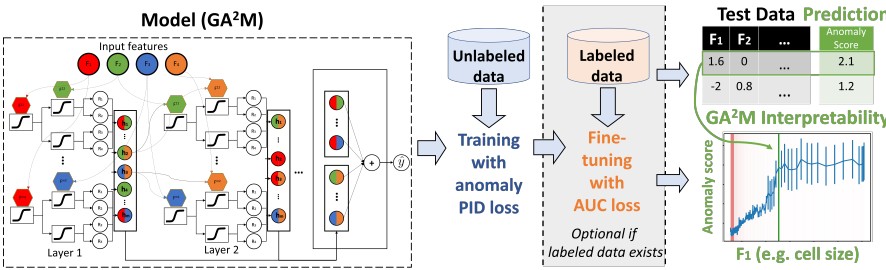

Figure 1: Overview of the proposed DIAD framework. During training, first an unsupervised AD model is fitted employing interpretable GA$^2$M models and PID loss with unlabeled data. Then, the trained unsupervised model is fined-tuned with a small amount of labeled data using a differentiable AUC loss. At inference, both the anomaly score and explanations are provided, based on the visualizations of top contributing features. The example sample in the figure is shown to have an anomaly score, explained by the cell size feature having high value.

(NodeGA$^2$M, Chang et al. (2021a)). Our design is based on the principle that PID scales to high-dimensional features and handles heterogeneous features well, while the differentiable GAM allows fine-tuning with labeled data and retain interpretability. In addition, PID requires clear-cut thresholds like trees which are provided by NodeGA$^2$M. While combining PID with NodeGA$^2$M, we introduce multiple methodological innovations, such as estimating and normalizing a sparsity metric as the anomaly scores, integrating a regularization for an inductive bias appropriate for AD, and using deep representation learning via fine-tuning with a differentiable AUC loss. The latter is crucial to take advantage of a small amount of labeled samples well and constitutes a more 'data-efficient' method compared to other AD approaches – e.g. DIAD improves from 87.1% to 89.4% AUC with 5 labeled anomalies compared to unsupervised AD. Overall, our innovations lead to strong empirical results – DIAD outperforms other alternatives significantly, both in unsupervised and semi-supervised settings. DIAD's outperformance is especially prominent on large-scale datasets containing heterogeneous features with complex relationships between them. In addition to accuracy gains, DIAD also provides a rationale on why an example is classified as anomalous using the GA$^2$M graphs, and insights on the impact of labeled data on the decision boundary, a novel explainability capability that provides both local and global understanding on the AD tasks.

Table 1: Comparison of AD approaches.

|  | Unlabeled data | Noisy features | Heterogenous features | Labeled data | Interpretability |
|---|---|---|---|---|---|
| PIDForest | ✓ | ✓ | ✓ | ✗ | ✗ |
| DAGMM | ✓ | ✗ | ✗ | ✗ | ✓ |
| GOAD | ✓ | ✓ | ✓ | ✗ | ✗ |
| Deep SAD | ✓ | ✓ | ✓ | ✓ | ✗ |
| SCAD | ✓ | ✗ | ✓ | ✗ | ✓ |
| DevNet | ✗ | ✓ | ✓ | ✓ | ✗ |
| DIAD (Ours) | ✓ | ✓ | ✓ | ✓ | ✓ |

## 2 RELATED WORK

**Overview of AD methods:** Table 1 summarizes representative AD works and compares to DIAD. AD methods for training with only normal data have been widely studied (Pang & Aggarwal, 2021b). Isolation Forest (IF) (Liu et al., 2008) grows decision trees randomly – the shallower the tree depth for a sample is, the more anomalous it is predicted. However, it shows performance degradation when feature dimensionality increases. Robust Random Cut Forest (RRCF, (Guha et al., 2016)) further improves IF by choosing features to split based on the range, but is sensitive to scale. PID-Forest (Gopalan et al., 2019) zooms on the features with large variances, for more robustness to noisy or irrelevant features. There are also AD methods based on generative approaches, that learn to reconstruct input features, and use the error of reconstructions or density to identify anomalies. Bergmann et al. (2019) employs auto-encoders for image data. DAGMM (Zong et al., 2018) first

learns an auto-encoder and then uses a Gaussian Mixture Model to estimate the density in the low-dimensional latent space. Since these are based on reconstructing input features, they may not be directly adapted to high-dimensional tabular data with noisy and heterogeneous features. Recently, methods with pseudo-tasks have been proposed as well. A major one is to predict geometric transformations (Golan & El-Yaniv, 2018; Bergman & Hoshen, 2019) and using prediction errors to detect anomalies. Qiu et al. (2021) shows improvements with a set of diverse transformations. CutPaste (Li et al., 2021) learns to classify images with replaced patches, combined with density estimation in the latent space. Lastly, several recent works focus on contrastive learning. Tack et al. (2020) learns to distinguish synthetic images from the original. Sohn et al. (2021) first learns a distribution-augmented contrastive representation and then uses a one-class classifier to identify anomalies. Self-Contrastive Anomaly Detection (SCAD) (Shenkar & Wolf, 2022) aims to distinguish in-window vs. out-of-window features by a sliding window and utilizes the error to identify anomalies.

**Explainable AD:** A few AD works focus on explainability as overviewed in Pang & Aggarwal (2021a). Vinh et al. (2016); Liu et al. (2020) explains anomalies using off-the-shelf detectors that might come with limitations as they are not fully designed for the AD task. Liznerski et al. (2021) proposes identifying anomalies with a one-class classifier (OCC) with an architecture such that each output unit corresponds to a receptive field in the input image. Kauffmann et al. (2020) also uses an OCC network but instead utilizes a saliency method for visualizations. These approaches can show the parts of images that lead to anomalies, however, their applicability is limited to image data, and they can not provide meaningful global explanations as GAMs.

**Semi-supervised AD:** Several works have been proposed for semi-supervised AD. Das et al. (2017), similar to ours, proposes a two-stage approach that first learns an IF on unlabeled data, and then updates the leaf weights of IF using labeled data. This approach can not update the tree structure learned in the first stage, which we show to be crucial for high performance (Sec. 6.4). Deep SAD (Ruff et al., 2019) extends deep OCC DSVDD (Ruff et al., 2018) to semi-supervised setting. However, this approach is not interpretable and underperforms unsupervised OCC-SVM on tabular data in their paper while DIAD outperforms it. DevNet (Pang et al., 2019b) formulates AD as a regression problem and achieves better sample complexity with limited labeled data. Yoon et al. (2020b) trains embeddings in self-supervised way (Kenton & Toutanova, 2019) with consistency loss (Sohn et al., 2020) and achieves state-of-the-art semi-supervised learning accuracy on tabular data.

## 3 PRELIMINARIES: $GA^2M$ AND $NODEGA^2M$

We first overview the NodeGA$^2$M model that we adopt in our framework, DIAD.

**GA$^2$M:** GAMs and GA$^2$Ms are designed to be interpretable with their functional forms only focusing on the 1st or 2nd order feature interactions and foregoing any 3rd-order or higher interactions. Specifically, given an input $x \in \mathbb{R}^D$, label $y$, a link function $g$ (e.g. $g$ is $\log(p/1-p)$ in binary classification), the main effects for $j^{(th)}$ feature $f_j$, and 2-way feature interactions $f_{jj'}$, the GA$^2$M models are expressed as:

$$\textbf{GA}^2\textbf{M}: g(y) = f_0 + \sum_{j=1}^{D} f_j(x_j) + \sum_{j=1}^{D} \sum_{j'>j} f_{jj'}(x_j, x_{j'}). \tag{1}$$

Unlike other high capacity models like DNNs that utilize all feature interactions, GA$^2$M are restricted to only lower-order, 2-way, interactions, so the impact of $f_j$ or $f_{jj'}$ can be visualized independently as a 1-D line plot and 2-D heatmap, providing a convenient way to gain insights behind the rationale of the model. On many real-world datasets, they can yield competitive accuracy, while providing simple explanations for humans to understand the rationale behind the model's decisions.

**NodeGA$^2$M:** NodeGA$^2$M (Chang et al., 2021a) is a differentiable extension of GA$^2$M which uses the neural trees to learn feature functions $f_j$ and $f_{jj'}$. Specifically, NodeGA$^2$M consists of $L$ layers where each layer has $m$ differentiable oblivious decision trees (ODT) whose outputs are combined with weighted superposition, yielding the model's final output. An ODT functions as a decision tree

with all nodes at the same depth sharing the same input features and thresholds, enabling parallel computation and better scaling. Specifically, an ODT of depth $C$ compares chosen $C$ input features to $C$ thresholds, and returns one of the $2^C$ possible options. $F^c$ chooses what features to split, thresholds $b^c$, and leaf weights $\boldsymbol{W} \in \mathbb{R}^{2^C}$, and its tree outputs $h(\boldsymbol{x})$ are given as:

$$h(\boldsymbol{x}) = \boldsymbol{W} \cdot \left( \begin{bmatrix} \mathbb{I}(F^1(\boldsymbol{x}) - b^1) \\ \mathbb{I}(b^1 - F^1(\boldsymbol{x})) \end{bmatrix} \otimes \cdots \otimes \begin{bmatrix} \mathbb{I}(F^C(\boldsymbol{x}) - b^C) \\ \mathbb{I}(b^C - F^C(\boldsymbol{x})) \end{bmatrix} \right), \tag{2}$$

where $\mathbb{I}$ is an indicator (step) function, $\otimes$ is the outer product and $\cdot$ is the inner product. To make ODT differentiable and in GA$^2$M form, Chang et al. (2021a) replaces the non-differentiable operations $F^c$ and $\mathbb{I}$ with differentiable relaxations via softmax and sigmoid-like functions. Each tree is allowed to interact with at most two features so there are no third- or higher-order interactions in the model. We provide more details in Appendix. B.

## 4 PARTIAL IDENTIFICATION AND SPARSITY AS THE ANOMALY SCORE

We consider the Partial Identification (PID) (Gopalan et al., 2019) as an AD objective given its benefits in minimizing the adversarial impact of noisy and heterogeneous features (e.g. mixture of multiple discrete and continuous types), particularly for tree-based models. By way of motivation, consider the data for all patients admitted to ICU – we might treat patients with blood pressure (BP) of 300 as anomalous, since very few people have more than 300 and the BP of 300 would be in such "sparse" feature space.

To formalize this intuition, we first introduce the concept of 'volume'. We consider the maximum and minimum value of each feature value and define the volume of a tree leaf as the product of the proportion of the splits within the minimum and maximum value. For example, assuming the maximum value of BP is 400 and minimum value is 0, the tree split of 'BP $\geq$ 300' has a volume 0.25. We define the sparsity $s_l$ of a tree leaf $l$ as the ratio between the volume of the leaf $V_l$ and the ratio of data in the leaf $D_l$ as $s_l = V_l / D_l$. Correspondingly, we propose treating higher sparsity as more anomalous – let's assume only less than 0.1% patients having values more than 300 and the volume of 'BP $\geq$ 300' being 0.25, then the anomalous level of such patient is the sparsity 0.25/0.1%. To learn effectively splitting of regions with high vs. low sparsity i.e. high v.s. low anomalousness, PIDForest (Gopalan et al., 2019) employs random forest with each tree maximizing the variance of sparsity across tree leafs to splits the space into a high (anomalous) and a low (normal) sparsity regions. Note that the expected sparsity weighted by the number of data samples in each leaf by definition is 1. Given each tree leaf $l$, the ratio of data in the leaf is $D_l$, sparsity $s_l$:

$$\mathbb{E}[s] = \sum_l [D_l s_l] = \sum_l \left[ D_l \frac{V_l}{D_l} \right] = \sum_l [V_l] = 1. \tag{3}$$

Therefore, maximizing the variance becomes equivalent to maximizing the second moment, as the first moment is 1:

$$\max \mathrm{Var}[s] = \max \sum_l D_l s_l^2 = \max \sum_l V_l^2 / D_l. \tag{4}$$

## 5 DIAD FRAMEWORK

In DIAD framework, we propose optimizing the tree structures of NodeGA$^2$M by gradients to maximize the PID objective – the variance of sparsity – meanwhile setting the leaf weights $\boldsymbol{W}$ in Eq. 2 as the sparsity of each leaf, so the final output is the sum of all sparsity values (anomalous levels) across trees. We overview the DIAD in Alg. 1. Details of DIAD framework are described below.

**Estimating PID**   The PID objective is based on estimating the ratio of volume $V_l$ and the ratio of data $D_l$ for each leaf $l$. However, exact calculation of the volume is not trivial in an efficient way for an oblivious decision tree as it requires complex rules extractions. Instead, we sample random points, uniformly in the input space, and count the number of the points that end up at each tree leaf as the empirical mean. More points in a leaf would indicate a larger volume. To avoid the zero count in the denominator, we employ Laplacian smoothing, adding a constant $\delta$ to each count.[1] Similarly, we estimate $D_l$ by counting the data ratio in each batch. We add $\delta$ to both $V_l$ and $D_l$.

---

[1] It is empirically observed to be important to set a large $\delta$, around 50-100, to encourage the models ignoring the tree leaves with fewer counts.

**Normalizing sparsity** The sparsity and thus the trees' outputs can have very large values up to 100s and can create challenges to gradient optimizations for the downstream layers of trees, and thus inferior performance in semi-supervised setting (Sec. 6.4). To address this, similar to batch normalization, we propose linearly scaling the estimated sparsity to be in [-1, 1] to normalize the tree outputs. We note that the linear scaling still preserves the ranking of the examples as the final score is a sum operation across all sparsity. Specifically, for each leaf $l$, the sparsity $s_l$ is:

$$\hat{s}_l = V_l/D_l, \quad s_l = 2\hat{s}_l/(\max_l \hat{s}_l - \min_l \hat{s}_l) - 1. \tag{5}$$

**Temperature annealing** We observe that the soft relaxation approach for tree splits in NodeGA$^2$M, EntMoid (which replace $\mathbb{I}$ in Eq. 2) does not perform well with the PID objective. We attribute this to Entmoid (similar to Sigmoid) being too smooth, yielding the resulting value similar across splits. Thus, we propose to make the split gradually from soft to hard operation during optimization:

$$\text{Entmoid}(\boldsymbol{x}/T) \to \mathbb{I} \text{ as optimization goes by linearly decresing } T \to 0 \tag{6}$$

**Updating leafs' weight** When updating the leaf weights $\boldsymbol{W}$ in Eq. 2 in each step to be sparsity, to stabilize its noisy estimation due to mini-batch and random sampling, we apply damping to the updates. Specifically, given the step $i$, sparsity $s_l^i$ for each leaf $l$, and the update rate $\gamma$ (we use $\gamma = 0.1$):

$$w_l^i = (1 - \gamma)w_l^{(i-1)} + \gamma s_l^i. \tag{7}$$

**Regularization** To encourage diverse trees, we introduce a novel regularization approach: per-tree dropout noise on the estimated momentum. We further restrict each tree to only split on $\rho\%$ of features randomly to promote diverse trees (see Appendix. I for details).

**Incorporating labeled data** At the second stage of fine-tuning using labeled data, we optimize the differentiable AUC loss (Yan et al., 2003; Das et al., 2017) which has been shown effective in imbalanced data setting. Note that we optimize

---

**Algorithm 1** DIAD training

**Input:** Mini-batch $\boldsymbol{X}$, tree model $\mathcal{M}$, smoothing $\delta$, $w_{tl}$ is an entry of the leaf weights matrix $\boldsymbol{W}$ (Eq. 2) for each tree $t$ and leaf $l$

---

$\boldsymbol{X}$ = MinMaxTransform($X$, min=$-1$, max=1)
$\boldsymbol{X_U} \sim U[-1, 1]$ {Data uniformly from [-1, 1]}
$E^{tl} = \mathcal{M}(\boldsymbol{X})$, $E_U^{tl} = \mathcal{M}(\boldsymbol{X_U})$ {Count how many data in each leaf $l$ of tree $t$ for $\boldsymbol{X}, \boldsymbol{X_U}$. See Alg. 2.}

$E^{tl} = E^{tl} + \delta$, $E_U^{tl} = E_U^{tl} + \delta$ {Smooth the counts}
$V_{tl} = \frac{E^{tl}}{\sum_{n'} E^{tl'}}$ {Volume ratio}
$D_{tl} = \frac{E_U^{tl}}{\sum_{n'} E_U^{tl'}}$ {Data ratio}
$M_{tl} = \frac{V_{tl}^2}{P_{tl}}$ {Second moments}
$L_M = -\sum_{t,l} M_{tl}$ {Maximize the second moments}
$\hat{s}_{tl} = V_{tl}/P_{tl}$ {Sparsity}
$s_{tl} = (\frac{2\hat{s}_{tl}}{(\max \hat{s}_{tl} - \min \hat{s}_{tl})} - 1)$ {Scale to [-1, 1] (Eq. 5)}
$w_{tl} = (1 - \gamma)w_{tl} + \gamma s_{tl}$ {Update tree weights – Eq. 7}
Optimize $L_M$ by Adam optimizer

---

both the tree structures and leaf weights of the DIAD. Specifically, given a randomly-sampled mini-batch of labeled positive/negative samples $X_P/X_N$, and the model $\mathcal{M}$, the objective is:

$$L_{PN} = 1/|X_P||X_N| \sum_{x_p \in X_P, x_n \in X_N} \max(\mathcal{M}(x_n) - \mathcal{M}(x_p), 0). \tag{8}$$

We show the benefit of this AUC loss compared to Binary Cross Entropy (BCE) in Sec. 6.4.

**Training data sampling** Similar to Pang et al. (2019a), we upsample the positive samples such that they have similar count with the negative samples, at each batch. We show the benefit of this over uniform sampling (see Sec. 6.4).

**Theoretical result** DIAD prioritizes splitting informative features rather than noise (Appendix C):

**Proposition 1** *Given uniform noise $x_n$ and non-uniform features $x_d$, DIAD prioritizes cutting $x_d$ over $x_n$ because the variance of sparsity of $x_d$ is larger than $x_n$ as sample size goes to infinity.*

## 6 EXPERIMENTS

We evaluate DIAD on various datasets, in both unsupervised and semi-supervised settings. Detailed settings and additional results are provided in the Appendix. Code will be released upon acceptance.

## 6.1 UNSUPERVISED ANOMALY DETECTION

We compare methods on 20 tabular datasets, including 14 datasets from Gopalan et al. (2019) and 6 larger datasets from Pang et al. (2019a).[2] We run and average results with 8 different random seeds.

**Baselines**  We compare DIAD with SCAD (Shenkar & Wolf, 2022), a recently-proposed deep learning based AD method, and other competitive methods including PIDForest (Gopalan et al., 2019), COPOD (Li et al., 2020), PCA, k-nearest neighbors (kNN), RRCF (Guha et al., 2016), LOF (Breunig et al., 2000) and OC-SVM (Schölkopf et al., 2001). To summarize performance across multiple datasets, we consider the averaged AUC (the higher, the better), as well as the average rank (the lower, the better) to avoid a few datasets dominating the results.

Table 2 shows that DIAD's performance is better or on par with others on most datasets. Compared to the PIDForest which has similar objectives, DIAD often underperforms on smaller datasets such as on Musk and Thyroid, but outperforms on larger datasets such as on Backdoor, Celeba, Census and Donors. To analyze the similarity of performances, Fig. 2 shows the Spearman correlation between rankings. DIAD is correlated with SCAD as they both perform better on larger datasets, attributed to better utilizing deep representation learning. PIDForest under-

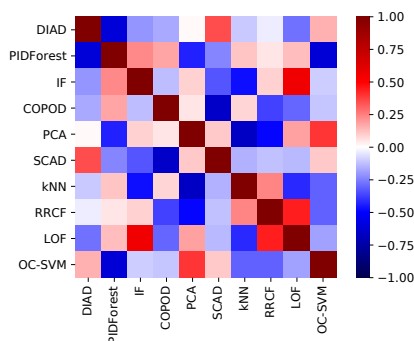

Figure 2: The Spearman correlation of methods' performance rankings. DIAD is correlated with SCAD as they both perform better in larger datasets. PID-Forest underperforms on larger datasets, and its correlation with DIAD is low, despite having similar objectives.

performs on larger datasets, and its correlation with DIAD is low despite having similar objectives.

Table 2: Unsupervised AD performance (% of AUC) on 18 datasets for DIAD and 9 baselines. Metrics with standard error overlapped with the best number are bolded. Methods without randomness don't have standard error. We show the number of samples (N) and the number of features (P), ordered by N.

| | DIAD | PIDForest | GIF | IF | COPOD | PCA | SCAD | kNN | RRCF | LOF | OC-SVM | N | P |
|---|---|---|---|---|---|---|---|---|---|---|---|---|---|
| Vowels | 78.3 ± 0.9 | 74.0 ± 1.0 | 79.0 ± 1.5 | 74.9 ± 2.5 | 49.6 | 60.6 | 90.8 ± 2.1 | **97.5** | 80.8 ± 0.3 | 5.7 | 77.8 | 1K | 12 |
| Siesmic | 72.2 ± 0.4 | 73.0 ± 0.3 | 53.3 ± 4.4 | 70.7 ± 0.2 | 72.7 | 68.2 | 65.3 ± 1.6 | **74.0** | 69.7 ± 1.0 | 44.7 | 60.1 | 3K | 15 |
| Musk | 90.8 ± 0.9 | **100.0** ± 0.0 | 93.2 ± 2.8 | **100.0** ± 0.0 | 94.6 | **100.0** | 93.3 ± 0.7 | 37.3 | 99.8 ± 0.1 | 58.4 | 57.3 | 3K | 166 |
| Satimage | **99.7** ± 0.0 | 98.2 ± 0.3 | 98.9 ± 0.6 | 99.3 ± 0.1 | 97.4 | 97.7 | 98.0 ± 1.3 | 93.6 | 99.2 ± 0.2 | 46.0 | 42.1 | 6K | 36 |
| Thyroid | 76.1 ± 2.5 | **88.2** ± 0.8 | 57.6 ± 6.0 | 81.4 ± 0.9 | 77.6 | 67.3 | 75.9 ± 2.2 | 75.1 | 74.0 ± 0.5 | 26.3 | 54.7 | 7K | 6 |
| A. T. | 78.3 ± 0.6 | **81.4** ± 0.6 | 56.4 ± 6.8 | 78.6 ± 0.6 | 78.0 | 79.2 | 79.3 ± 0.7 | 63.4 | 69.9 ± 0.4 | 43.7 | 67.0 | 7K | 10 |
| NYC | 57.3 ± 0.9 | 57.2 ± 0.6 | 49.0 ± 3.2 | 55.3 ± 1.0 | 56.4 | 51.1 | 64.5 ± 0.9 | **69.7** | 54.4 ± 0.5 | 32.9 | 50.0 | 10K | 10 |
| Mammo. | 85.0 ± 0.3 | 84.8 ± 0.4 | 82.5 ± 0.3 | 85.7 ± 0.5 | **90.5** | 88.6 | 69.8 ± 2.7 | 83.9 | 83.2 ± 0.2 | 28.0 | 87.2 | 11K | 6 |
| CPU | 91.9 ± 0.2 | 93.2 ± 0.1 | 78.1 ± 0.9 | 91.6 ± 0.2 | **93.9** | 85.8 | 87.5 ± 0.3 | 72.4 | 78.6 ± 0.3 | 44.0 | 79.4 | 18K | 10 |
| M. T. | 81.2 ± 0.2 | 81.6 ± 0.3 | 73.9 ± 12.9 | 82.7 ± 0.5 | 80.9 | **83.4** | 81.8 ± 0.4 | 75.9 | 74.7 ± 0.4 | 49.9 | 79.6 | 23K | 10 |
| Campaign | 71.0 ± 0.8 | **78.6** ± 0.8 | 64.1 ± 3.9 | 70.4 ± 1.9 | **78.3** | 73.4 | 72.0 ± 0.5 | 72.0 | 65.5 ± 0.3 | 46.3 | 66.7 | 41K | 62 |
| smtp | 86.8 ± 0.5 | **91.9** ± 0.2 | 76.7 ± 5.3 | 90.5 ± 0.7 | 91.2 | 82.3 | 82.2 ± 2.0 | 89.5 | 88.9 ± 2.3 | 9.5 | 84.1 | 95K | 3 |
| Backdoor | **91.1** ± 2.5 | 74.2 ± 2.6 | 66.9 ± 8.4 | 74.8 ± 4.1 | 78.9 | 88.7 | **91.8** ± 0.6 | 66.8 | 75.4 ± 0.7 | 28.6 | 86.1 | 95K | 196 |
| Celeba | **77.2** ± 1.9 | 67.1 ± 4.8 | 61.6 ± 6.0 | 70.3 ± 0.8 | 75.1 | **78.6** | 75.4 ± 2.6 | 56.7 | 61.7 ± 0.3 | 56.3 | 68.5 | 203K | 39 |
| Fraud | **95.7** ± 0.2 | 94.7 ± 0.3 | 80.4 ± 0.8 | 94.8 ± 0.1 | 94.7 | 95.2 | **95.5** ± 0.2 | 93.4 | 87.5 ± 0.4 | 52.5 | 94.8 | 285K | 29 |
| Census | 65.6 ± 2.1 | 53.4 ± 8.1 | 58.8 ± 2.5 | 61.9 ± 1.9 | 66.1 | 66.1 | 58.4 ± 0.9 | 64.6 | 55.7 ± 0.1 | 45.0 | 53.4 | 299K | 500 |
| http | 99.3 ± 0.1 | 99.2 ± 0.2 | 91.1 ± 7.0 | **100.0** ± 0.0 | 99.2 | 99.6 | 99.3 ± 0.1 | 23.1 | 98.4 ± 0.2 | 64.7 | 99.4 | 567K | 3 |
| Donors | **87.7** ± 6.2 | 61.1 ± 1.3 | 80.3 ± 18.2 | 78.3 ± 0.7 | 81.5 | 82.9 | 65.5 ± 11.8 | 61.2 | 64.1 ± 0.0 | 40.2 | 70.2 | 619K | 10 |
| Average | **82.5** | 80.7 | 71.2 | 81.2 | 81.0 | 80.5 | 80.3 | 70.6 | 76.8 | 40.2 | 71.0 | - | - |
| Rank | **3.6** | 4.4 | 6.3 | 4.0 | 4.2 | 4.2 | 4.7 | 6.6 | 6.7 | 9.8 | 6.8 | - | - |

Next, we show the robustness of AD methods with additional noisy features. We follow the experimental settings from Gopalan et al. (2019) to include 50 additional noisy features which are randomly sampled from $[-1, 1]$ on Thyroid and Mammography datasets, and their noisy versions. Table. 3 shows that the performance of DIAD is more robust with additional noisy fea-

---

[2]We did not use all 30 datasets in ODDS used in SCAD (Shenkar & Wolf, 2022) because some are small or overlap with datasets from (Gopalan et al., 2019).

tures (76.1→71.1, 85.0→83.1), while others show significant performance degradation. On Thyroid (noise), SCAD decreases from 75.9→49.5, KNN from 75.1→49.5, and COPOD from 77.6→60.5.

Table 3: Unsupervised AD performance (% of AUC) with additional 50 noisy features for DIAD and 9 baselines. We find both DIAD and OC-SVM deteriorate around 2-3% while other methods deteriorate 7-17% on average.

|  | DIAD | PIDForest | GIF | IF | COPOD | PCA | SCAD | kNN | RRCF | LOF | OC-SVM |
|---|---|---|---|---|---|---|---|---|---|---|---|
| Thyroid | $76.1_{\pm 2.5}$ | $88.2_{\pm 0.8}$ | $57.6_{\pm 6.0}$ | $81.4_{\pm 0.9}$ | 77.6 | 67.3 | $75.9_{\pm 2.2}$ | 75.1 | $74.0_{\pm 0.5}$ | 26.3 | 54.7 |
| Thyroid (noise) | $71.1_{\pm 1.2}$ | $76.0_{\pm 2.9}$ | $49.4_{\pm 1.2}$ | $64.4_{\pm 1.6}$ | 60.5 | 61.4 | $49.5_{\pm 1.6}$ | 49.5 | $53.6_{\pm 1.1}$ | 50.8 | 49.4 |
| Mammography | $85.0_{\pm 0.3}$ | $84.8_{\pm 0.4}$ | $82.5_{\pm 0.3}$ | $85.7_{\pm 0.5}$ | 90.5 | 88.6 | $69.8_{\pm 2.7}$ | 83.9 | $83.2_{\pm 0.2}$ | 28.0 | 87.2 |
| Mammography (noise) | $83.1_{\pm 0.4}$ | $82.0_{\pm 2.2}$ | $72.7_{\pm 5.4}$ | $71.4_{\pm 2.0}$ | 72.4 | 76.8 | $69.4_{\pm 2.4}$ | 81.7 | $79.1_{\pm 0.7}$ | 37.2 | 87.2 |
| Average ↓ | **3.5** | 7.5 | 9.1 | 15.6 | 17.6 | 8.9 | 13.4 | 13.9 | 12.2 | -16.8 | **2.7** |

## 6.2 SEMI-SUPERVISED ANOMALY DETECTION

Next, we focus on the semi-supervised setting and show DIAD can take advantage of small amount of labeled data in a superior way.

We divide the data into 64%-16%-20% train-val-test splits and within the training set, we consider that only a small part of data is labeled. We assume the existence of labels for a small subset of the training set (5, 15, 30, 60 or 120 positives and the corresponding negatives to have the same anomaly ratio).

The validation set is used for model selection and we report the averaged performances evaluated on 10 disjoint data splits. We compare

Table 4: Performance in semi-supervised AD setting. We show the average % of AUC across 15 datasets with varying number of anomalies.

| No. Anomalies | 0 | 5 | 15 | 30 | 60 | 120 |
|---|---|---|---|---|---|---|
| DIAD | **87.1** | **89.4** | **90.0** | **90.4** | **89.4** | **91.0** |
| DIAD w/o PT | - | 86.2 | 87.6 | 88.3 | 87.2 | 88.8 |
| CST | - | 85.3 | 86.5 | 87.1 | 86.6 | 88.8 |
| DevNet | - | 83.0 | 84.8 | 85.4 | 83.9 | 85.4 |

to 3 baselines: (1) **DIAD w/o PT**: optimized with labeled data without the first AD pre-training stage, (2) **CST**: VIME with consistency loss (Yoon et al., 2020a) which regularizes the model to make similar predictions between unlabeled data under dropout noise injection, (3) **DevNet** (Pang et al., 2019a): a state-of-the-art semi-supervised AD approach. Further details are provided in Appendix. I.2.

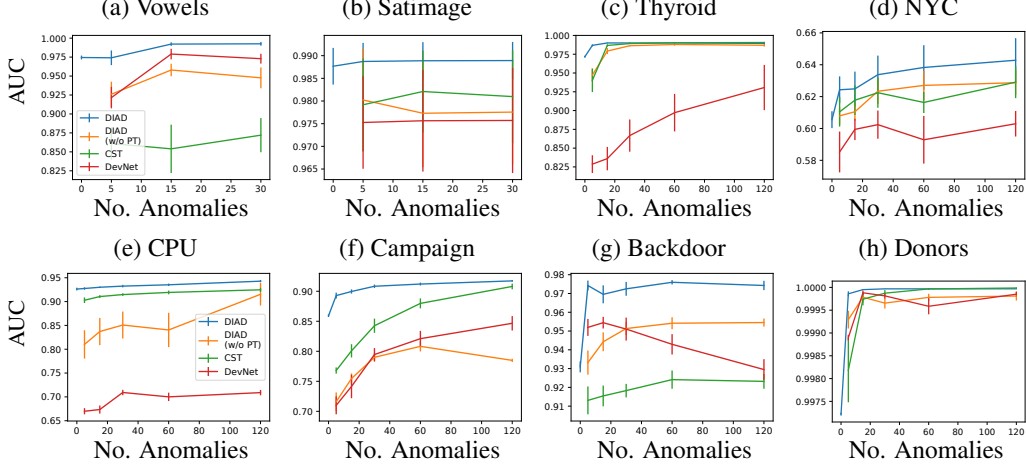

Figure 3: Semi-supervised AD performance on 8 tabular datasets (out of 15) with varying number of anomalies. Our method 'DIAD' (blue) outperforms other semi-supervised baselines. Summarized results can be found in Table. 4. Remaining plots with 7 tabular datasets are provided in Appendix. G.

Fig. 3 shows the AUC across 8 of 15 datasets (the rest can be found in Appendix. G). The proposed version of DIAD (blue) outperforms DIAD without the first stage pre-training (orange) consistently on 14 of 15 datasets (except Census), which demonstrates that learning the PID objective with unlabeled data improves the performance. Second, neither the VIME with consistency loss (green) or DevNet (red) always improves the performance compared to the supervised setting. Table 4 shows the average AUC of all methods in semi-supervised AD. Overall, DIAD outperforms all baselines and shows improvements over the unlabeled setting. In Appendix. E, we show similar results in average ranks metric rather than AUC.

### 6.3 QUALITATIVE ANALYSES ON DIAD EXPLANATIONS

**Explaining anomalous data**   DIAD provides value to the users by providing insights on why a sample is predicted as anomalous. We demonstrate this by focusing on Mammography dataset and showing the explanations obtained by DIAD for anomalous samples. The task is to detect breast cancer from radiological scans, specifically the presence of clusters of microcalcifications that appear bright on a mammogram. The 11k images are segmented and preprocessed using standard computer vision pipelines and 6 image-related features are extracted, including the area of the cell, constrast, and noise. Fig. 4 shows the data samples that are predicted to be the most anomalous, and the rationale behind DIAD on feature contributes more for the anomaly score. The unusually-high 'Contrast' (Fig. 4(a)) is a major factor in the way image differs from other samples. The unusually high noise (Fig. 4(b)) and 'Large area' (Fig. 4(c)) are other ones. In addition, Fig. 4(d) shows 2-way interactions and the insights by it on why the sample is anomalous. The sample has 'middle area' and 'middle gray level', which constitute a rare combination in the dataset. Overall, these visualizations shed light into which features are the most important ones for a sample being considered as anomalous, and how the value of the features affect the anomaly likelihood.

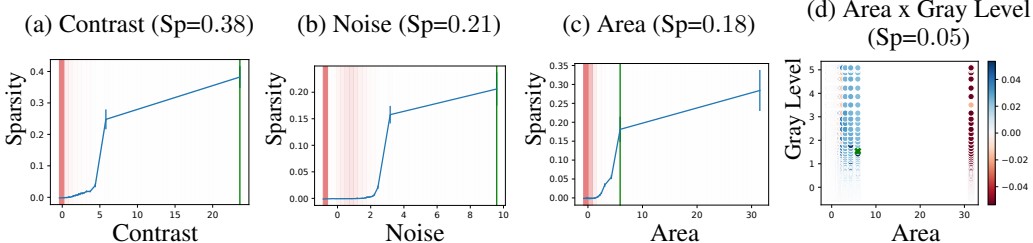

Figure 4: Explanations of the most anomalous samples on the Mammography dataset. We show the top 4 contributing features ordered by the sparsity (Sp) value (anomalous levels) of our model, with 3 features (a-c) and 1 two-way interaction (d). (a-c) x-axis is the feature value, and y-axis is the model's predicted sparsity (higher sparsity represents more likelihood of being anomalous). Model's predicted sparsity is shown as the blue line. The red backgrounds indicate the data density and the green line indicates the value of the most anomalous sample with Sp as its sparsity. The model finds it anomalous as it has high Contrast, Noise and Area, different from values that a majority of other samples have. (d) x-axis is the Area and y-axis is the Gray Level with color indicating the sparsity (blue/red indicates anomalous/normal). The green dot is the value of the data that has 0.05 sparsity.

**Qualitative analyses on the impact of fine-tuning with labeled data**   Fig. 5 visualizes how predictions change before and after fine-tuning with labeled samples on Donors dataset. Donors dataset consists of 620k educational proposals for K12 level with 10 features. The anomalies are defined as the top 5% ranked proposals as outstanding. We show visualizations before and after fine-tuning. Figs. 5 a & b show that both 'Great Chat' and 'Great Messages Proportion' increase in magnitude after fine-tuning, indicating that the sparsity (as a signal of anomaly likelihood) of these is consistent with the labels. Conversely, Figs. 5 c & d show the opposite trend after fine-tuning. The sparsity definition treats the values with less density as more anomalous – in this case *'Fully Funded'=0* is treated as more anomalous. In fact, 'Fully Funded' is a well-known indicator of outstanding proposals, so after fine-tuning, the model learns that *'Fully Funded'=1* in fact contributes to a higher anomaly score. This underlines the importance of incorporating labeled data to improve AD accuracy. Appendix. H shows another visualization in Thyroid dataset.

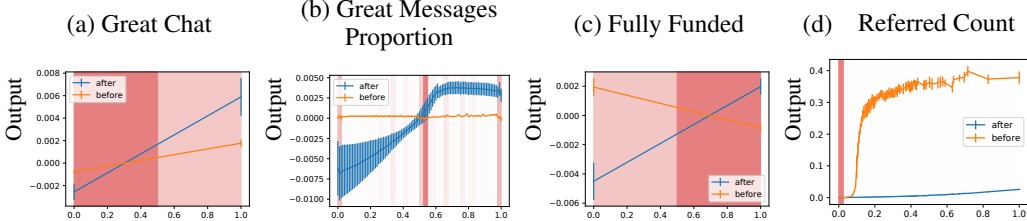

Figure 5: AD decision visualizations on the Donors dataset before (orange) and after (blue) fine-tuning with the labeled samples. Here the darker/lighter red in the background indicates high/low data density and thus less/more anomalous. In (a, b) we show the two features that after fine-tuning (blue) the magnitude increases which shows the labels agree with the notion of data sparsity learned before fine-tuning (orange). In (c, d) the label information disagrees with the notion of sparsity; thus, the magnitude changes or decreases after the fine-tuning.

## 6.4 ABLATION AND SENSITIVITY ANALYSIS

Table 5: Ablation study for semi-supervised AD.

| No. Anomalies | 5 | 15 | 30 | 60 | 120 |
|---|---|---|---|---|---|
| DIAD | **89.4** | **90.0** | **90.4** | **89.4** | **91.0** |
| Only AUC | 88.9 | 89.4 | 90.0 | 89.1 | 90.7 |
| Only BCE | 88.8 | 89.3 | 89.4 | 88.3 | 89.2 |
| Unnormalized sparsity | 84.1 | 85.6 | 85.7 | 84.2 | 85.6 |
| No upsampling | 88.6 | 89.1 | 89.4 | 88.5 | 90.1 |
| Only finetune leaf weights | 84.8 | 85.7 | 86.6 | 85.7 | 88.3 |

Table 6: Semi-supervised AD performance with 25% of the validation data.

| 25% val data (4% of total data) | | | | | |
|---|---|---|---|---|---|
| No. Anomalies | 5 | 15 | 30 | 60 | 120 |
| DIAD | **89.0** | **89.3** | **89.7** | **89.1** | **90.4** |
| DIAD w/o PT | 85.4 | 87.1 | 86.9 | 86.4 | 87.9 |
| CST | 83.9 | 84.9 | 85.7 | 85.6 | 88.2 |
| DevNet | 82.0 | 83.4 | 84.4 | 82.0 | 84.6 |

To analyze the major constituents of DIAD, we perform ablation analyses, presented in Table 5. We show that fine-tuning with AUC outperforms BCE. Sparsity normalization plays an important role in fine-tuning, since sparsity could have values up to $10^4$ which negatively affect fine-tuning. Upsampling the positive samples also contributesto performance improvements. Finally, to compare with Das et al. (2017) which updates the leaf weights of a trained IF (Liu et al., 2008) to incorporate labeled data, we propose a variant that only fine-tunes the leaf weights using labeled data in the second stage without changing the tree structure learned in the first stage, which substantially decreases the performance. Using differentiable trees that update both leaf structures and weights is also shown to be important.

In practice we might not have a large validation dataset, as in Sec. 6.2, thus, it would be valuable to show strong performance with a small validation dataset. In Table 6, we reduce the validation dataset size to only 4% of the labeled data and find DIAD still consistently outperforms others. Additional results can be found in Appendix. D. We also perform sensitivity analysis in Appendix. F that varies hyperparameters in the unsupervised AD benchmarks. Our method is quite stable with less than 2% differences across a variety of hyperparameters on many different datasets.

## 7 DISCUSSIONS AND CONCLUSIONS

As unsupervised AD methods rely on approximate objectives to discover anomalies such as reconstruction loss, predicting geometric transformations, or contrastive learning, the objectives inevitably would not align with labels on some datasets, as inferred from the performance ranking fluctuations across datasets. This motivates for incorporating labeled data to boost performance and interpretability to find out whether the model could be trusted and whether the approximate objective aligns with the human-defined anomalies. In this paper, we consider the general PNU setting, with both positive (P) and negative (N) samples available and we propose a novel interpretable AD framework, DIAD. Our framework consists multiple novel contributions that are key for highly accurate and interpretable AD: we introduce a novel way to estimate and normalize sparsity, modify the architecture by temperature annealing, propose a novel regularization for improved generalization, and introduce semi-supervised AD via supervised fine-tuning of the unsupervised learnt representations. These play a crucial role in pushing the state-of-the-art in both unsupervised and semi-supervised AD benchmarks. Furthermore, DIAD provides unique interpretability capabilities, crucial in high-stakes applications such as in finance or healthcare.

REPRODUCIBILITY STATEMENT

We provide our hyperparameters in Appendix I. We will provide code upon paper acceptance.

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

## A  PSEUDO CODE FOR SOFT DIFFERENTIABLE OBLIVIOUS TREES - ALG. 2

Here, we show the pseudo code of differentiable trees.

---

**Algorithm 2** Soft decision tree training

---

**Input:** Mini-batch $\boldsymbol{X} \in \mathbb{R}^{B \times D}$, Temperature $T_1, T_2$ $(T_1, T_2 \to 0)$
**Symbols:** Tree Depth $C$, Entmoid $\sigma$
**Trainable Parameters**: Feature selection logits $\boldsymbol{G}^1, \boldsymbol{G}^2 \in \mathbb{R}^D$, split Thresholds $\boldsymbol{b} \in \mathbb{R}^C$, split slope $\boldsymbol{S} \in \mathbb{R}^C$,

---

$\boldsymbol{G} = [\boldsymbol{G}^1, \boldsymbol{G}^2, \boldsymbol{G}^1, ...]^T \in \mathbb{R}^{D \times C}$ {Alternating $\boldsymbol{G}^1, \boldsymbol{G}^2$ so only 2 chosen features per tree}
$G = \boldsymbol{X} \cdot \text{EntMax}(\boldsymbol{F}/T_1, \dim=0) \in \mathbb{R}^{B \times C}$ {Weighted sum to soft-select features with temperature $T_1$}
**for** $c = 1$ **to** $C$ **do**
$\quad H^c = \sigma(\frac{(\boldsymbol{G^c} - b^c)}{S^c \cdot T_2})$ {Soft binary split of the feature value with temperature $T_2$}
**end for**
$\boldsymbol{e} = \left( \begin{bmatrix} H^1 \\ (1 - H^1) \end{bmatrix} \otimes \cdots \otimes \begin{bmatrix} (H^C) \\ (1 - H^C) \end{bmatrix} \right) \in \mathbb{R}^{B \times 2^C}$ {Soft counts between $[0, 1]$ per leaf}
$E = \text{sum}(\boldsymbol{e}, \dim=0) \in \mathbb{R}^{2^C}$ {Sum across batch to get total counts per leaf}
**Return:** $E$ count

---

## B  DETAILS OF MAKING TREE OPERATIONS DIFFERENTIABLE

Both $F^c(\boldsymbol{x})$ and I would prevent differentiability. To address this, $F^c(\boldsymbol{x})$ is replaced with a weighted sum of features with temperature annealing that makes it gradually sharper:

$$F^c(\boldsymbol{x}) = \sum_{j=1}^{D} x_j \text{entmax}_\alpha(\boldsymbol{G}^c/T)_j, \quad T \to 0, \tag{9}$$

where $\boldsymbol{G}^c \in \mathbb{R}^D$ is a trainable vector per layer $c$ per tree, and entmax$_\alpha$ (Peters et al., 2019) is the entmax normalization, as the sparse version of softmax whose output sum equals to 1. As $T \to 0$, the output of entmax gradually becomes one-hot and $F^c(\boldsymbol{x})$ picks only one feature. Similarly, the step function $\mathbb{I}$ is replaced with entmoid, which is a sparse sigmoid with outputs between 0 and 1. Differentiability of all operations (entmax, entmoid, outer/inner products), render ODT differentiable to optimize parameters $\boldsymbol{W}$, $b^c$ and $\boldsymbol{G}^c$ (Chang et al., 2021a).

## C  PROOF OF PROPOSITION 1

**Proposition 1**  *Given uniform noise $x_n$ and non-uniform features $x_d$, DIAD prioritizes cutting $x_d$ over $x_n$ because the variance of sparsity of $x_d$ is larger than $x_n$ as sample size goes to infinity.*

Here, we show that the variance of sparsity of uniform features would go to 0 under large sample sizes. Without loss of generality, we assume that the decision tree has a single cut in $l \in (0, 1)$ in a uniform feature $x_n \in [0, 1]$, and we denote the sparsity of the left segment as $s_1$ and the right as $s_2$. The sparsity $s_1$ is defined as $\frac{V_1}{D_l}$ where the $V_1 = l$ is the volume, and the $D_l$ is the data ratio i.e. $D_1 = \frac{c_1}{n}$ where $c_1$ is the counts of samples in segment 1 between 0 and $l$, and $n$ is the total samples. Since $x_n$ is a uniform feature, the counts $c_1$ become a Binomial distribution with $n$ samples and probability $l$:

$$c_1 \sim \text{Bern}(n, l), c_2 \sim \text{Bern}(n, 1 - l).$$

As $n \to \infty$, $D_1 \to l$ because $\mathbb{E}[D_l] = \frac{\mathbb{E}[c_l]}{n} = l$ and $Var[D_1] = \frac{Var[c_l]}{n^2} = \frac{l(1-l)}{n} \to 0$. Therefore, as number of examples grow, the sparsity $s_1 = \frac{V_1}{D_l} \to \frac{l}{l} = 1$. Similarly, $s_2 \to 1$. For any uniform noise, since both sparsity $s_1, s_2$ converges to 1 as $n \to \infty$ no matter where the cut is, the variance of sparsity converges to 0. Thus, the objective of DIAD which maximizes the variance of sparsity would prefer splitting other non-uniform features since there is no gain in variance of sparsity when splitting on the uniform noise.

## D  SEMI-SUPERVISED AD RESULTS WITH SMALLER VALIDATION SET

When we have a small set of labeled data, how should we split it between the train and validation datasets when optimizing semi-supervised methods? In Sec. 6.2 we use 64%-16%-20% for train-val-test splits, and 16% of validation set could be too large for some real-world settings. Does our method still outperform others under a smaller validation set?

To answer this, we experiment with a much smaller validation set with only 50% and 25% of original validation set (i.e. 8% and 4% of total datasets). In Table 7, we show the average AD performance across 15 datasets with varying size of validation data. With decreasing validation size all methods decrease the performance slightly, our method still consistently outperforms others.

Table 7: Summary of Semi-supervised AD performances with varying size of validation set (4%, 8% and 16% of total datasets). We show the average % of AUC across 15 datasets with varying number of anomalies. Our method DIAD still outperforms others consistently.

| | 25% val data (4% of total data) | | | | | 50% val data (8% of total data) | | | | |
|---|---|---|---|---|---|---|---|---|---|---|
| No. Anomalies | 5 | 15 | 30 | 60 | 120 | 5 | 15 | 30 | 60 | 120 |
| DIAD w/o PT | 85.4 | 87.1 | 86.9 | 86.4 | 87.9 | 85.7 | 86.9 | 88.0 | 86.9 | 87.5 |
| DIAD | **89.0** | **89.3** | **89.7** | **89.1** | **90.4** | **89.2** | **89.7** | **90.0** | **89.2** | **90.6** |
| CST | 83.9 | 84.9 | 85.7 | 85.6 | 88.2 | 84.2 | 85.7 | 85.8 | 86.2 | 87.9 |
| DevNet | 82.0 | 83.4 | 84.4 | 82.0 | 84.6 | 83.0 | 85.0 | 85.5 | 83.6 | 85.5 |

| | 100% val data (16% of total data) | | | | | - | | | | |
|---|---|---|---|---|---|---|---|---|---|---|
| No. Anomalies | 5 | 15 | 30 | 60 | 120 | - | | | | |
| DIAD w/o PT | 86.2 | 87.6 | 88.3 | 87.2 | 88.8 | - | | | | |
| DIAD | **89.4** | **90.0** | **90.4** | **89.4** | **91.0** | - | | | | |
| CST | 85.3 | 86.5 | 87.1 | 86.6 | 88.8 | - | | | | |
| DevNet | 83.0 | 84.8 | 85.4 | 83.9 | 85.4 | - | | | | |

## E  THE AVERAGE RANK PERFORMANCE OF SEMI-SUPERVISED AD RESULTS

The average AUC for semi-supervised AD results (Table 4) might not represent the entire picture, so we provide the average ranks as well in Table 8. Our method still consistently outperforms other methods.

Table 8: Average ranks of AUC across 15 datasets in the Semi-supervised AD result.

| No. Anomalies | 5 | 15 | 30 | 60 | 120 |
|---|---|---|---|---|---|
| DIAD | **1.3** | **1.3** | **1.3** | **1.3** | **1.2** |
| DIAD w/o PT | 2.3 | 2.6 | 2.6 | 2.5 | 2.9 |
| CST | 3.2 | 3.1 | 3.1 | 3.0 | 2.7 |
| DevNet | 3.1 | 3.0 | 3.1 | 3.2 | 3.2 |

## F  SENSITIVITY ANALYSIS

We perform sensitivity analyses from our default hyperparameter in the unsupervised AD benchmarks. We exclude Census, NYC taxi, SMTP, and HTTP datasets since some hyperparameters can not be run, resulting in total 14 datasets each with 4 different random seeds. In Fig. 6, besides showing the average of all datasets (blue), we also group datasets by their sample sizes into 3 groups: (1) $N > 10^5$ (Orange, 3 datasets), (2) $10^5 > N > 10^4$ (green, 5 datasets), and (4) $N < 10^4$ (red, 6 datasets). Overall, DIAD shows quite stable performance between 82-84 *except* when (c) No. trees= 50 and (h) smoothing $\leq 10$. We also find that 3 hyperparameters: (a) batch size, (b) No. Layers, and (d) Tree depth that the large group (orange) has an opposite trend than the small group (red). Large datasets yield better results with smaller batch sizes, larger layers, and shallower tree depths.

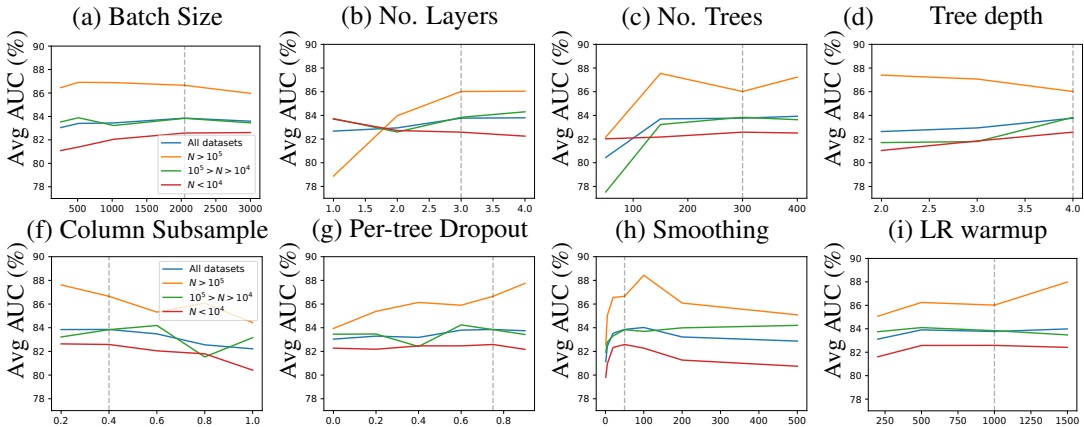

Figure 6: Sensitivity analysis. Y-axis shows the average AUC across 14 datasets, and X-axis shows the varying hyperparameters. The dashed line is the default hyperparameter. We show 4 groups: (1) All datasets (Blue), (2) $N > 10^5$ (Orange), (3) $10^5 > N > 10^4$ (green), and (4) $N < 10^4$ (red).

## G  SEMI-SUPERVISED AD FIGURES

We experiment with 15 datasets and measure the performanceunder a different number of anomalies. We split the dataset into 64-16-20 train-val-test splits and run 10 times to report the mean and standard error. We show the performance in Fig. 7.

## H  MORE VISUAL EXPLANATIONS

We show another example of DIAD explanations on the "Celeba" dataset. Celeba consists of 200K pictures of celebrities and annotated with 40 attributes including "Bald", "Hair", "Mastache", "Attractive" etc. We train the DIAD on these 40 sets of attributes and treat the "Bald" attribute as outliers since it accounts for only 3% of all celebrities. Here we show the most anomalous example deemed by the DIAD in Fig. 8. The top 4 contributing factors are shown in (a-d), showing Gray Hair, Mustache, Receding Hairline, and Rosy Cheeks are very anomalous in the data. We also show the top 4 interactions in (e-h), indicating the combination of Rosy Cheeks with Mustache, Goatee, Necktie and Side Burns are even more anomalous deemed by DIAD.

We also show the least anomalous example deemed by DIAD in the Celeba dataset in Fig. 9. The lack of "Receding Hairline", "Rosy Cheeks", "Pale Skin", and "Gray Hair" are pretty common and thus DIAD outputs a negative normalized sparsity value.

We show another example of DIAD explanations on the "Backdoor" dataset. It consists of 95K samples and 196 features that record the backdoor network attacks with the attacks as anomalies against the 'normal' class, which is extracted from the UNSW-NB 15 data set. In Fig. 10, we show two most anomalous examples deemd by DIAD. The Fig. 10(a-c) shows the top 3 contributing factors for one example and the "protocol=HMP" solely determines its high abnormity since the rest of the two features have only little sparsity. A user can thus decide if he wants to trust such explanation and finds out if such protocol is indeed anomalous. The Fig. 10(d-f) shows the top 3 contributing factors for the other example and both the "protocol=ICMP" and "state=ECO" contributes to its high sparsity (1.5, and 1.2 respectively). And other features are relatively quite normal.

We further show another case study of DIAD explanations on "Thyroid" datasets before and after fine-tuning to further demonstrate the discrepancy between unsupervised AD objective and labeled anomalies. Thyroid datasets contains 7200 samples with 6 features and 500 of labels are labeled as "hyperfunctioning". In Fig. 11, we visualize 4 attributes: (a) T3, (b) T4U, (c) TBG, and (d) TT4. And the dark red in the backgrounds indicates the high data density by bucketizing the x-axis into 64 bins and counts the number of examples for each bin. First, in T3 feature, before fine-tuning (orange) the model predicts a higher anomaly for values above 0 since they have little density and have mostly white region. After fine-tuning on the labeled data (blue), the model further strengthens

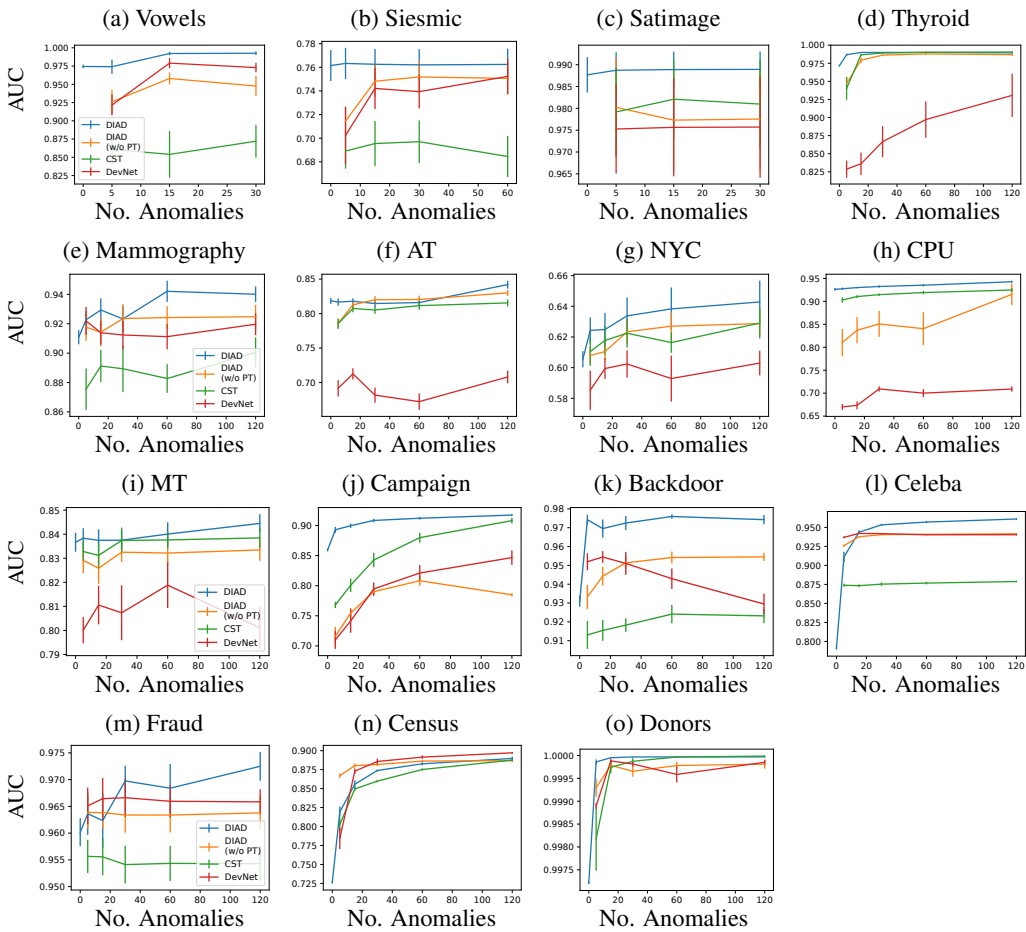

Figure 7: Semi-supervised AD performance on 8 tabular datasets (out of 15) with varying number of anomalies. Our method 'DIAD' (blue) outperforms other semi-supervised baselines. Table. 4 summarizes the comparisons.

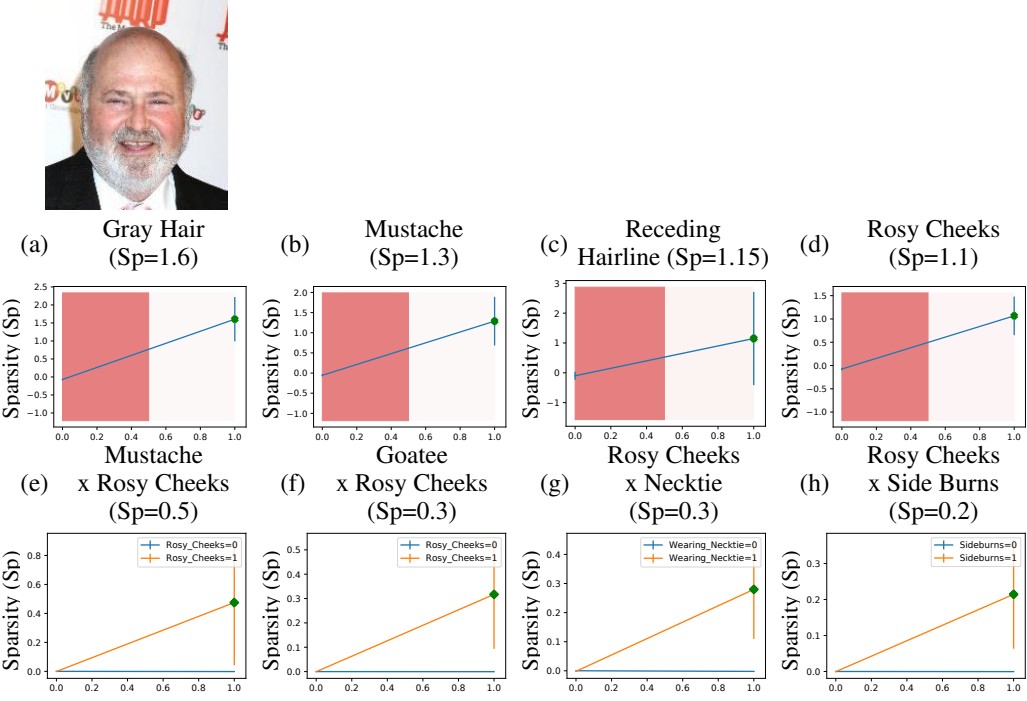

Figure 8: DIAD decision making visualizations of the most anomalous image of the CelebA dataset. The top 4 contributing mains are shown in (a-d) where the green dots are the image's attributes and the blue line is the model's prediction. This celebrity has gray hair, Mustache, receding hairline, and rosy cheeks which make DIAD predict him as very anomalous in the dataset. The top 4 2-way interactions are shown in (e-h) where the combination of the Rosy Cheeks with Mustache (e), Goatee (f), Necktie (g), and Side Burns (h) make him even more anomalous.

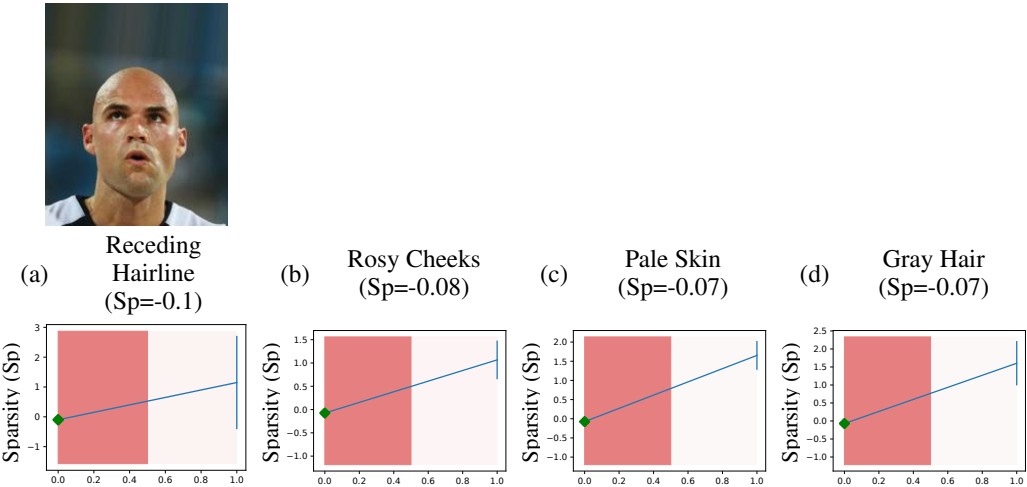

Figure 9: DIAD decision making visualizations of the least anomalous celebrity in the CelebA dataset, showing its least 4 anomalous features. The lack of "receding hariline", "rosy cheeks", "pale skin", and "gray hair" make DIAD deem him as the most normal subject indicated by the negative sparsity.

its belief that values bigger than 0 are anomalous. However, in T4U, TBG and TT4 features, before fine-tuning (orange) model indicates higher values are more anomalous because it has larger volume and relatively small data density (white). But after fine-tuning (blue) on the labels the model

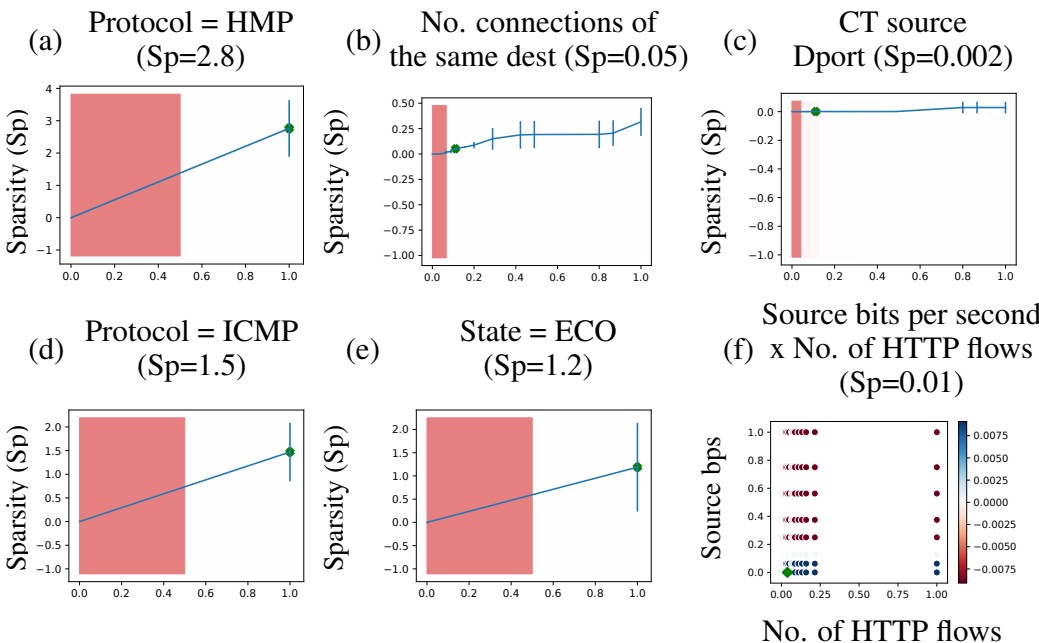

Figure 10: DIAD decision making visualizations of 2 most anomalous examples in the Backdoor dataset. (a-c) shows the top 3 contributing factors of one example and the "protocol=HMP" is solely responsible for its abnormity prediction. (d-f) shows another example that both "protocol=ICMP" and "State=ECO" are both contributing to large abnormity value.

moves to an opposite direction that the smaller feature value is more anomalous. This shows that the used unsupervised anomaly objective, PID, is in conflict with the human-defined anomalies in these features.

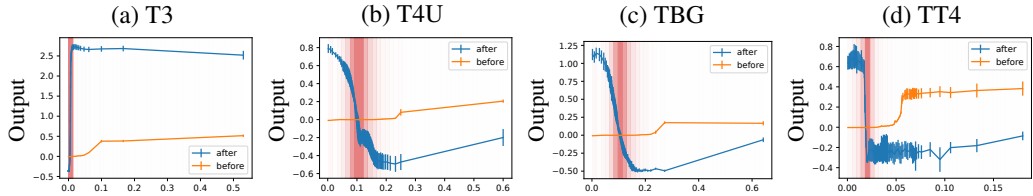

Figure 11: AD decision making visualizations on the Thyroid dataset before (orange) and after (blue) fine-tuning on the labeled samples. Here the darker/lighter red in the background indicates high/low data density and thus less/more anomalous. In (a) T3, the labeled information agrees with the anomaly specified in PID, so after fine-tuning the magnitude increases. In (b, c, d) the label information disagrees with the anomalous levels specified in PID especially when the value are small; thus, the magnitude changes or decreases after the fine-tuning.

We show more experimental results to see how methods perform under noise injection in Table 9, following the procedures described in Sec. 6 and Table 3. In additional to Thyroid and Mammograph, we further compare with Siesmic, Campaign, and Fraud. We find that overall DIAD and OC-SVM only deteriorates around 1-2% while others can deteriorate up to 3-11% on average, showing DIAD's superiority of noise resistance.

# I  HYPERPARAMETERS

Here we list the hyperparameters we use for both unsupervised and semi-supervised experiments.

Table 9: Unsupervised AD performance (% of AUC) with additional 50 noisy features for DIAD and 9 baselines. We find both DIAD and OC-SVM deteriorate on average 1-2% while other methods deteriorate 3-11% on average. We ignore the average of LOF since most of the ROC are below 50%.

| | DIAD | PIDForest | GIF | IF | COPOD | PCA | SCAD | kNN | RRCF | LOF | OC-SVM |
|---|---|---|---|---|---|---|---|---|---|---|---|
| Thyroid | $76.1_{\pm2.5}$ | $88.2_{\pm0.8}$ | $57.6_{\pm6.0}$ | $81.4_{\pm0.9}$ | 77.6 | 67.3 | $75.9_{\pm2.2}$ | 75.1 | $74.0_{\pm0.5}$ | 26.3 | 54.7 |
| Thyroid (noise) | $71.1_{\pm1.2}$ | $76.0_{\pm2.9}$ | $49.4_{\pm1.2}$ | $64.4_{\pm1.6}$ | 60.5 | 61.4 | $49.5_{\pm1.6}$ | 49.5 | $53.6_{\pm1.1}$ | 50.8 | 49.4 |
| Mammography | $85.0_{\pm0.3}$ | $84.8_{\pm0.4}$ | $82.5_{\pm0.3}$ | $85.7_{\pm0.5}$ | 90.5 | 88.6 | $69.8_{\pm2.7}$ | 83.9 | $83.2_{\pm0.2}$ | 28.0 | 87.2 |
| Mammography (noise) | $83.1_{\pm0.4}$ | $82.0_{\pm2.2}$ | $72.7_{\pm5.4}$ | $71.4_{\pm2.0}$ | 72.4 | 76.8 | $69.4_{\pm2.4}$ | 81.7 | $79.1_{\pm0.7}$ | 37.2 | 87.2 |
| Siesmic | $72.2_{\pm0.4}$ | $73.0_{\pm0.3}$ | $53.3_{\pm4.4}$ | $70.7_{\pm0.2}$ | 72.7 | 68.2 | $65.3_{\pm1.6}$ | 74.0 | $69.7_{\pm1.0}$ | 44.7 | 58.9 |
| Siesmic (noise) | $71.8_{\pm0.1}$ | $72.1_{\pm0.7}$ | $56.9_{\pm2.2}$ | $66.0_{\pm1.2}$ | 68.3 | 65.5 | $58.0_{\pm1.8}$ | 74.0 | $69.7_{\pm0.3}$ | 44.7 | 60.1 |
| Campaign | $71.0_{\pm0.8}$ | $78.6_{\pm0.8}$ | $64.1_{\pm3.9}$ | $70.4_{\pm1.9}$ | 78.3 | 73.4 | $72.0_{\pm0.5}$ | 72.0 | $65.5_{\pm0.3}$ | 46.3 | 66.7 |
| Campaign (noise) | $70.6_{\pm0.5}$ | $72.1_{\pm1.3}$ | $63.7_{\pm0.3}$ | $67.6_{\pm0.8}$ | 71.1 | 72.5 | $71.6_{\pm0.5}$ | 67.6 | $59.3_{\pm0.0}$ | 40.3 | 64.5 |
| Fraud | $95.7_{\pm0.2}$ | $94.7_{\pm0.3}$ | $80.4_{\pm0.8}$ | $94.8_{\pm0.1}$ | 94.7 | 95.2 | $95.5_{\pm0.2}$ | 93.4 | $87.5_{\pm0.4}$ | 52.5 | 94.8 |
| Fraud (noise) | $95.6_{\pm0.0}$ | $94.6_{\pm0.1}$ | $51.1_{\pm0.6}$ | $93.4_{\pm0.4}$ | 93.8 | 95.1 | $92.7_{\pm0.7}$ | 78.6 | $61.6_{\pm0.4}$ | 21.3 | 92.9 |
| Average $\downarrow$ | **1.6** | 3.3 | 9.5 | 8.0 | 9.5 | 4.3 | 7.5 | 9.4 | 11.3 | - | 1.9 |

## I.1 UNSUPERVISED AD

Since it's hard to tune hyperparameters in unsupervised setting, for fair comparisons, we use all baselines with default hyperparameters. Here we list the default hyperparameter for DIAD in Table 10. Here we explain each specific hyperparameter:

- Batch size: the sample size of mini-batch.
- LR: learning rate.
- $\gamma$: the hyperparameter used to update the sparsity in each leaf (Eq. 7).
- Steps: the total number of training steps. We find 2000 works well across our datasets.
- LR warmup steps: we do the learning rate warmup (Goyal et al., 2017) that linearly increases the learning rate from 0 to 1e-3 in the first 1000 steps.
- Smoothing $\delta$: the smoothing count for our volume and data ratio estimation.
- Per tree dropout: the dropout noise we use for the update of each tree.
- Arch: we adopt the GAMAtt architecture form the NodeGAM (Chang et al., 2021a).
- No. layer: the number of layers of trees.
- No. trees: the number of trees per layer.
- Additional tree dimension: the dimension of the tree's output. If more than 0, it appends an additional dimension in the output of each tree.
- Tree depth: the depth of tree.
- Dim Attention: since we use the GAMAtt architecture, this determines the size of the attention embedding. We find tuning more than 32 will lead to insufficient memory in our GPU, and in general 8-16 works well.
- Column subsample ($\rho$): this controls how many proportion of features a tree can operate on.
- Temp annealing steps (K), Min Temp: these control how fast the temperature linearly decays from 1 to the minimum temperature (0.1) in K steps. After K training steps, the entmax or entmoid become max or step functions in the model.

## I.2 SEMI-SUPERVISED AD

We adopt 2-stage training. In the 1st stage, we optimize the AD objective and select a best model by the validation set performance under the random search. Then in the 2nd stage, we search the following hyperparameters with No. anomalies=120 to choose the best hyperparamter, and later run through the rest of 5, 15, 30, and 60 anomalies to report the performances.

- Learning Rate: [5e-3, 2e-3, 1e-3, 5e-4]

Table 10: Default hyperparameter used in the unsupervised AD benchmarks.

| Batch Size | LR | $\gamma$ | Steps | LR warmup steps | Smoothing | Per tree Dropout | Arch |
|---|---|---|---|---|---|---|---|
| 2048 | 0.001 | 0.1 | 2000 | 1000 | 50 | 0.75 | GAMAtt |

| No. layers | No. trees | Addi. tree dim | Tree depth | Dim Attention | Column Subsample ($\rho$) | Temp annealing steps (K) | Min Temp |
|---|---|---|---|---|---|---|---|
| 3 | 300 | 1 | 4 | 12 | 0.4 | 1000 | 0.1 |

- Loss: ['AUC', 'BCE'].

Then, for each baseline we use the same architecture but tune the hyperparameters:

- CST: the overall loss is calcualted as follows (Eq. 7, 8, 9 in (Yoon et al., 2020a)):

$$L_{final} = L_s + \beta L_u$$

The supervised loss $L_s$ is:

$$L_s = \mathbb{E}_{(\boldsymbol{x},y)\sim P_{XY}}[l_{BCE}(y, f(\boldsymbol{x}))]$$

The consistency loss $L_u$ is:

$$L_u = \mathbb{E}_{\boldsymbol{x}\sim P_X, \boldsymbol{m}\sim p_{\boldsymbol{m}}, \hat{\boldsymbol{x}}\sim g_m(\boldsymbol{x},\boldsymbol{m})}[(f_e(\hat{\boldsymbol{x}}) - f_e\boldsymbol{x})^2]$$

where the $g_m(\boldsymbol{x}, \boldsymbol{m})$ is to use dropout mask $\boldsymbol{m}$ to remove features and impute it with the marginal feature distribution, and the masks are sampled $K$ times. Since the accuracy is quite stable across different $\beta$, and when $K \geq 20$ (Fig. 10, (Yoon et al., 2020a)), we select $\beta = 1$ and $K = 20$, and search the dropout rate $p_{\boldsymbol{m}}$ from [0.05, 0.1, 0.2, 0.35, 0.5, 0.7] and the learning rate [2e-3, 1e-3].

- DevNet: they first randomly sample 5000 Gaussian samples with 0 mean and 1 standard deviation and calculate the mean $u_R$ and standard deviation $\sigma_R$:

$$u_R = \frac{1}{l}\sum_{i=1}^{l} r_i \quad \text{where} \ r_i \sim N(0, 1),$$

$$\sigma_R = \text{standard deviation of } \{r_1, r_2..., r_{5000}\}.$$

Then they calculate the loss (Eq. 6, 7 in (Pang et al., 2019a)):

$$L = (1 - y)|dev(x)| + y\max(0, a - dev(\boldsymbol{x})) \quad \text{where} \ dev(\boldsymbol{x}) = \frac{\phi(\boldsymbol{x}) - u_R}{\sigma_R}.$$

The $\phi$ is the deep neural network and the $a$ is set to 5. In short, they try to increase the output of anomalies ($y = 1$) to be bigger than $a$ and let the output of normal data ($y = 0$) to be close to 0. We tune learning rates from [2e-3, 1e-3, 5e-4] for DevNet.

