# OpenReview forum: "Data-Efficient and Interpretable Tabular Anomaly Detection"
_ICLR.cc/2023/Conference — Submitted to ICLR 2023_

### Official Review · Reviewer_235S · 2022-10-17

**Confidence:** 3
**Correctness:** 3
**Technical Novelty And Significance:** 2
**Empirical Novelty And Significance:** 3
**Recommendation:** 5

**Clarity, Quality, Novelty And Reproducibility:**

Overall, the paper is not very hard to follow as long as the readers have knowledge about NodeGAM and PID. By combining these two approaches, DIAD can achieve interpretable anomaly detection on tabular data, which is somewhat novel. The authors promise to provide code upon paper acceptance.

**Strength And Weaknesses:**

Strength
1. The paper targets an important research problem, interpretable anomaly detection.
2. The proposed framework leverages the existing techniques and makes sense.
3. Detailed evaluation is conducted to show the performance of anomaly detection.


Weakness:
1. In the unsupervised setting, the proposed DIAD can achieve better or comparable performance on large datasets, but may not be as good as baselines when the datasets only have small amounts of samples. Do the authors have any insight into such observations?

2. My major concern is in the interpretation part. How to achieve interpretable anomaly detection based on DIAD is not very clear to me. For example, given an anomaly, do we explain the result based on the ranking of features evaluated by the sparsity values or based on the feature-wise function? Meanwhile, as Mammo. only consists of 6 features, I am also curious about the interpretability of the proposed model on high-dimensional data. I believe for low-dimensional data, it is not very challenging for domain experts to exam the anomalies.

**Summary Of The Paper:**

This paper develops an interpretable anomaly detection framework, DIAD,  by combining the NodeGAM structure with the PID objective. The NodeGAM provides an interpretable mechanism, and PID can be trained for anomaly detection. Overall, the framework makes sense to me. The paper includes detailed experimental evaluation by comparing with several state-of-the-art baselines on multiple datasets.

**Summary Of The Review:**

I feel that the proposed approach has some merits, but I am also not very sure whether the combination of NodeGAM and PID is trivial or challenging. Meanwhile, the term "interpretable anomaly detection" in the title really draws my eye, but the evaluation in this part is weak in my view. I understand currently there is no golden metric to evaluate the interpretability, but maybe showing more cases could be more convincing, just like the papers in the CV area.

I also have one more question, which may be outside the scope of this paper: If the purpose is to achieve sample-wise interpretation (explaining the anomalies), could we apply other post-hoc interpretation techniques to interpret PIDForest?

---

> ### Author Response · Authors · 2022-11-19
> **Thank you (1/2)**
>
> > "In the unsupervised setting, the proposed DIAD can achieve better or comparable performance on large datasets, but may not be as good as baselines when the datasets only have small amounts of samples. Do the authors have any insight into such observations?”
>
>
> Large-scale datasets can benefit from high-capacity models more. We observe that on small-scale datasets, simple AD methods often suffice and the high-capacity ones such as the architecture in DIAD would not be able to show its advantages. This is consistent in the supervised learning field, for example the TabNet, a deep learning architecture in the tabular data, can only outperform the GBDT in larger datasets. We also emphasize that despite the number of samples being large, the amount of collected labels can still be very small in most real-world settings, and DIAD provides significant advantages in them.
>
>
>
> > “"My major concern is in the interpretation part. How to achieve interpretable anomaly detection based on DIAD is not very clear to me. For example, given an anomaly, do we explain the result based on the ranking of features evaluated by the sparsity values or based on the feature-wise function?”
>
>
>
> The short answer is both. To clarify, the sparsity values are just the y-value read from the feature-wise function with corresponding x-value of the sample.
>
> The key benefit of GAM is that it only models only 1st and 2nd order feature interactions that allows visualizations, as it is hard to visualize a 3rd order interactions in a 4-d graph. Therefore, we can visualize each feature-wise function in a graph to understand how model predicts differently across the entire range of feature values to observe the trend. For example, this helps humans understand how increasing or decreasing the feature values affects the model’s output. This has been shown to be useful to detect fairness, missing-not-at-random issues, and important spurious patterns. See (1), (2) for example.
>
> On the other hand, we can also observe which features give an example a high score (sparsity) and use it as the ranking of the features, which is like other local explanation methods that provide per-feature importance per example.
>
> To clarify it more clearly and demonstrate its benefit, we show more visualizations in Appendix H.
>
>
> (1) Caruana, Rich, et al. "Intelligible models for healthcare: Predicting pneumonia risk and hospital 30-day readmission." Proceedings of the 21th ACM SIGKDD international conference on knowledge discovery and data mining. 2015.
>
> (2) Chang, Chun-Hao, et al. "How interpretable and trustworthy are gams?." Proceedings of the 27th ACM SIGKDD Conference on Knowledge Discovery & Data Mining. 2021.
>
>
>
> > "I feel that the proposed approach has some merits, but I am also not very sure whether the combination of NodeGAM and PID is trivial or challenging.”
>
>
>
> It’s surprisingly challenging to increase the performance of the DIAD model during our developments. The machinery we introduced including estimation, smoothing, normalizations all contribute to the final strong performance. For example, the smoothing value should be large like 100-200, unlike traditional wisdom around 1.
>
> While combining PID with NodeGAM, we introduce multiple methodological innovations, such as estimating and normalizing a sparsity metric as the anomaly scores, integrating a regularization for an inductive bias appropriate for AD, and using deep representation learning via fine-tuning with a differentiable AUC loss. Overall, our innovations lead to strong empirical results -- DIAD outperforms other alternatives significantly, both in unsupervised and semi-supervised settings.
>
>
>
> > Meanwhile, the term "interpretable anomaly detection" in the title really draws my eye, but the evaluation in this part is weak in my view. I understand currently there is no golden metric to evaluate the interpretability, but maybe showing more cases could be more convincing, just like the papers in the CV area.
>
> Thank you, and thank you for awaring that there is no golden metric in the interpretability. We thus have included more visualization examples in Appendix H as you suggested.

---

> > ### Author Response · Authors · 2022-11-19
> > **Thank you (2/2)**
> >
> >
> >
> > > "I also have one more question, which may be outside the scope of this paper: If the purpose is to achieve sample-wise interpretation (explaining the anomalies), could we apply other post-hoc interpretation techniques to interpret PIDForest?
> >
> >
> >
> > It’s indeed possible in principle but it comes with many limitations compared to DIAD. First, the post-hoc interpretation technique such as SHAP to derive per-feature importance cannot capture the full picture of the model that utilizes higher-order feature interactions, while the visualizations and feature importance from DIAD are exact since we limit it to be only 1 or 2-way interactions.
> >
> > Also, the graphs of the feature-functions can be very helpful for users to understand the global behavior of the model, and gain insights of the dataset. For example, the Donors dataset we find that the labeled anomalies are sometimes against the unsupervised AD objective in some features. Thus, it is helpful to either design a better unsupervised objective, for example not optimizing the PID under some features, or just remove the features from the dataset.

---

### Official Review · Reviewer_qGLb · 2022-10-19

**Confidence:** 4
**Correctness:** 4
**Technical Novelty And Significance:** 3
**Empirical Novelty And Significance:** 3
**Recommendation:** 6

**Clarity, Quality, Novelty And Reproducibility:**

The paper is well written and easy to read and follow. The approach is novel. The reproducibility is rather high, but could be improved at places (see above).

**Strength And Weaknesses:**

Strengths:
- Innovative approach based on generative additive models.
- The model is explainable.
- There is a little technical contribution (Prop 1) creating some insight into the method's inner workings.

Weaknesses:
- Section 6.1. on unsupervised AD experiments has some problems in terms of the chosen datasets and baselines: [1] determined [2] as the overall best performing deep AD method on tabular data. However, the method [2] is missing in the comparison here. On the 14 datasets from Gopalan et al., the proposed approach shows inferior performance, except for SatImage, where a tiny improvement of 0.4% is achieved. Out of the 9 datasets from Pang et al., only 6 have been selected. To avoid dataset selection bias, all 9 should be analyzed. Besides NeuTraL, also DSVDD - a common baseline - is missing in the comparison.
- In the noise experiments, only two datasets have been analyzed, which is insufficient to draw conclusions.
- Section 6.2 on semi-supervised AD: These experiments look better. The approach achieves higher AUROC than the Devnet baseline. However, the original Devnet paper reported on AUPRC, which is why I cannot confirm that the present results are consistent with their analysis. The semi-supervised DSVDD baseline is missing.



[1] Maxime Alvarez, Jean-Charles Verdier, D'Jeff K. Nkashama, Marc Frappier, Pierre-Martin Tardif, Froduald Kabanza: A Revealing Large-Scale Evaluation of Unsupervised Anomaly Detection Algorithms. https://arxiv.org/abs/2204.09825
[2] Chen Qiu, Timo Pfrommer, Marius Kloft, Stephan Mandt, Maja Rudolph Proceedings of the 38th International Conference on Machine Learning, PMLR 139:8703-8714, 2021.

**Summary Of The Paper:**

The paper proposes a new interpretable method for unsupervised or semi-supervised anomaly detection on tabular data, in the presence of noisy or unlabeled data.

**Summary Of The Review:**

In summary, the authors propose an innovative approach to AD on tabular data. The experimental evaluation shows good promise, but is not entirely convincing.

---

> ### Author Response · Authors · 2022-11-19
> **Thank you for your favorable reviews**
>
> > "Section 6.1. on unsupervised AD experiments has some problems in terms of the chosen datasets and baselines: ... On the 14 datasets from Gopalan et al., the proposed approach shows inferior performance, except for SatImage, where a tiny improvement of 0.4% is achieved. “
>
>
>
> We believe the reviewer may have missed that our method DIAD also outperforms PIDForest by 4.3% in the first dataset “Vowels”. Indeed, we find our method does not perform well in the datasets of PIDForest which are usually in the small data regime. We think it’s because large-scale datasets can benefit from high-capacity models more. We observe that on small-scale datasets, simple AD methods often suffice and the high-capacity ones such as the architecture in DIAD would not be able to show its advantages. This is consistent in the supervised learning field, for example the TabNet, a deep learning architecture in the tabular data, can only outperform the GBDT in larger datasets.
>
>
>
> > “Out of the 9 datasets from Pang et al., only 6 have been selected. To avoid dataset selection bias, all 9 should be analyzed.”
>
>
>
> Thank you for pointing this out. We were not aware of this because in the code release of the paper only 7 datasets are included, where one of the datasets Thyroid overlaps with the datasets from PIDForest, and thus we only mention 6. We further investigate if we can reproduce the datasets but then we realize the authors use the “sparse random projection” to reduce their dimensionality to 1000. Since these methods involve randomness and we are not very familiar with them, we feel we cannot reproduce the results faithfully. We hope it alleviates your concern about dataset picking.
>
>
>
> > "[1] determined [2] as the overall best performing deep AD method on tabular data. However, the method [2] is missing in the comparison here. Besides NeuTraL, also DSVDD - a common baseline - is missing in the comparison.
>
> Thank you for your suggestions. We find that NeuTraL has very different hyperparameters per dataset so it seems that NeuTral needs to tune its performance on the labeled data which is a different setting. However, in our setting we assume there is no labeled data available so hyperparameter tuning is prohibited, and it is difficult to do hyperparameter tuning for each dataset to reproduce their methods.
>
>
>
> For DSVDD, as stated in our related work, the DSVDD performs poorly in the original paper on the tabular data which consistently and substantially underperforms its non-deep-learning counterpart OCC-SVM, so we instead only compare with OCC-SVM in our Table 1 and show DIAD is indeed much better. Please see Appendix B of the original paper (1). Also, similar to NeuTraL, the DSVDD does not have a good default hyperparameter which makes it difficult to run in the unsupervised setting with so many hyperparameters to set. Finally, the SCAD method which we show DIAD outperforms heavily compares with an improved version of DSVDD (DROCC) in their paper and shows its superiority.
>
>
>
> We also include a new baseline, Generalized Isolation Forest (GIF), as suggested by R1 and show our method is still better than GIF.
>
> Ruff, Lukas, et al. "Deep semi-supervised anomaly detection." arXiv preprint arXiv:1906.02694 (2019).
>
>
>
> > In the noise experiments, only two datasets have been analyzed, which is insufficient to draw conclusions.
>
> Thank you. We only compare these two datasets by following the PIDForest paper. Thus we included a larger study by analyzing total 5 datasets with varying sample sizes in Appendix I. Our method DIAD is still the most noise-resistant method among all others.
>
>
>
> > Section 6.2 on semi-supervised AD: These experiments look better. The approach achieves higher AUROC than the Devnet baseline. However, the original Devnet paper reported on AUPRC, which is why I cannot confirm that the present results are consistent with their analysis. The semi-supervised DSVDD baseline is missing.
>
>
> In the Table 1 and Table 2, the Devnet paper provides both AUPRC and AUROC, although Devnet only reports AUPRC under varying labeled sample sizes. We choose AUROC because the PIDForest and the DevNet both use this metric and we want to be consistent across experiments.
>
> We omit semi-supervised DSVDD because the original paper finds poorer performance in tabular data compared to other traditional baselines in Appendix B. For example, comparing their method (Deep SAD) with the OCC-SVM in Table 6, the OCC-SVM consistently outperforms or is on par with Deep SAD. Also, in the DevNet paper the DSVDD is also compared and performs poorer.

---

### Official Review · Reviewer_1y6T · 2022-10-25

**Confidence:** 5
**Clarity, Quality, Novelty And Reproducibility:** The paper clarity and technical novel…
**Correctness:** 2
**Technical Novelty And Significance:** 1
**Empirical Novelty And Significance:** Not applicable
**Recommendation:** 5

**Strength And Weaknesses:**

The work aims to tackle both the interpretability and the capability of utilizing a few labeled anomaly examples in anomaly detection. It has the following strengths:
- Both of the interpretability and the sample-efficient capability are important to different ML tasks, including anomaly detection. The paper addresses both of these two important aspects.
- The presented method can work in both unsupervised and semi-supervised settings. Limited AD methods have such a property.
- Experiments on 18 datasets show the effectiveness of the method against popular unsupervised/semi supervised methods

Some issues that may require further investigation include:
- The technical novelty of the method is weak. The methods like PID were used in anomaly detection, and the AUC loss is a widely used loss function and it does not bring much improvement over BCE. The full method seems to be a combination of these widely used components.
- The method is weak in the unsupervised anomaly detection setting, which does not show clear improvement over the closely related competing methods like PIDForest and IF. There are also other more effective improved IF methods that tackle the studied issues in the paper, such as [a-c] and references therein. It would be more convincing to review these studies and possibly include them in the comparison.
- In terms of the semi-supervised setting, the proposed method involves two stages of training, unsupervised training and fine-tuning. I wonder whether the competing methods like CST and DevNet are trained from scratch, or only fine-tuned in a similar way as the proposed method. Questions here include how much the contribution is it from the unsupervised training in the semi-supervised setting? Does such a training strategy also help improve other existing semi-supervised methods?
- The work argues a two-way feature interaction-based anomaly explanation, but the presented result in Figure 4 is only the explanation of individual features. Figure 4(d) is a two-way interaction example, but the results indicate only individual features, e.g., the Area feature, are sufficient for the detection and interpretation.
- The interpretation results are limited to qualitative ones, and do not involve comparison to other methods, and thus, they are not convincing enough.
- There exist a large number of gradient backpropagation-based methods for anomaly explanation, especially on image data. This type of methods is applicable to tabular data, too. The authors are suggested to elaborate more based on more related work in the discussion in Related Work.

**References**
- [a] "Sparse modeling-based sequential ensemble learning for effective outlier detection in high-dimensional numeric data." In Proceedings of the AAAI Conference on Artificial Intelligence, vol. 32, no. 1. 2018.
- [b] "Extended isolation forest." IEEE Transactions on Knowledge and Data Engineering 33, no. 4 (2019): 1479-1489.
- [c] "Generalized isolation forest for anomaly detection." Pattern Recognition Letters 149 (2021): 109-119.

**Summary Of The Paper:**

The paper introduces a new anomaly detection method, namely DIAD, that uses Partial Identification (PID) as an objective to perform anomaly detection optimization with a tree structure of an existing generalized additive model. It is also flexible to use an additional loss function, such as AUC optimization or BCE loss, to utilize auxiliary labeled anomaly data in the semi-supervised setting. Due to the use of GAM model structures, it also offers a two-way feature interaction based explanation of detected anomalies. The effectiveness of the method is evaluated on 18 tabular datasets.

**Summary Of The Review:**

Considering both the pros and cons above, the paper is at the borderline and towards the reject side.

---

> ### Author Response · Authors · 2022-11-19
> **Thank you; Please see replies below**
>
>
>
> > The technical novelty of the method is weak. The methods like PID were used in anomaly detection, and the AUC loss is a widely used loss function and it does not bring much improvement over BCE. The full method seems to be a combination of these widely used components.
>
> We believe the components may have been published but introducing key changes to make them work and have superior performance is non-trivial at all. Without our introduced changes such as sparsity smoothing, estimation, architecture changes, and sparsity normalization, the naïve combinations will perform much worse. Also, we believe it is an understudied area that introduces interpretability into the AD setting that unprecedentedly provides both global and local explanations. DIAD is also able to further utilize the labeled anomalies in the AD setting where very few methods can do. We propose DIAD with non-trivial changes to the existing components and achieve superior performance in both unsupervised and semi-supervised AD.
>
>
> > "The method is weak in the unsupervised anomaly detection setting, which does not show clear improvement over the closely related competing methods like PIDForest and IF. There are also other more effective improved IF methods that tackle the studied issues in the paper, such as [a-c] and references therein. It would be more convincing to review these studies and possibly include them in the comparison.
>
> Although we believe we already compare very comprehensively across the literature, we further compare to the Generalized Isolation Forest in Table 2 and show our method outperforms it. We also believe our result of 82.5% v.s. 80.7% of IF itself is a large improvement that might have significant real-world impact in problems from many industries such as Manufacturing or Healthcare or Finance.
>
>
> > The work argues a two-way feature interaction-based anomaly explanation, but the presented result in Figure 4 is only the explanation of individual features. Figure 4(d) is a two-way interaction example, but the results indicate only individual features, e.g., the Area feature, are sufficient for the detection and interpretation.
>
>
> Thank you for your suggestions. We thus include another visualization case study on the CelebA dataset in the Appendix H, and show several top 2-way interactions plots in Fig. 9 (e-f).
>
>
> In Fig. 4, the top 3 contributing factors are univariate so we show them first in Fig. 4(a-c). In Fig. 4(d), it cannot just use Area to detect anomalies because within the same area (the same x-value) there could be both positive (blue) and negative (red) sparsity value depending on different “Gray Level” (y-axis), since it is a 2-way feature interaction.
>
>
>
> > The interpretation results are limited to qualitative ones, and do not involve comparison to other methods, and thus, they are not convincing enough.
>
>
>
> Our method is the first of its kind to provide global interpretability in AD literature. That is, there is no other interpretable method that provides such global interpretability, so it’s unclear to us how to provide quantitative comparisons. The interpretable methods listed in Table 1 provide only local explanations which are not directly quantitatively comparable to global ones like us. Also, in the interpretability area, it’s often unclear what the ground truth explanation should look like to quantitatively compare, and even harder for different forms of explanations. For example, how should one compare the Shapley value, the rule lists and a decision tree?
>
> To respond to this, we provide more qualitative examples as in the CV field in Appendix H that show the utilities of the explanations.
>
>
> > There exist a large number of gradient backpropagation-based methods for anomaly explanation, especially on image data. This type of methods is applicable to tabular data, too. The authors are suggested to elaborate more based on more related work in the discussion in Related Work.
>
>
> We believe we have already illustrated several works that utilize these backpropagation-based methods in the related work under the “Explainable AD” section, and have discussed the key differences between theirs to ours. If there is anything specific in your mind, please let us know.

---

### Official Review · Reviewer_XPd4 · 2022-10-25

**Confidence:** 3
**Correctness:** 3
**Technical Novelty And Significance:** 2
**Empirical Novelty And Significance:** 3
**Recommendation:** 5

**Clarity, Quality, Novelty And Reproducibility:**

The paper is clear, only two comments on presentation:
- Section 6.3 of the results (part “Explaining anomalous data): this section is not completely convincing. Are you extracting something that other methods can not extract?
Moreover, plots in Fig 4 are too small, it is very difficult to get colors, lines and in general the content

- References: in general there are many citations to arxiv papers, also for old papers. Please try to avoid this whenever it is possible. There are also some duplicated entries:
- Pang and Aggarwal 21
- Pang, Shen, van den Hengel 19
- Yoon et al 2020

For what concerns novelty:
Estimation of the PID in section 5. Maybe authors can consider also the work by Goix and colleagues, which extends Isolation Forests by optimizing a criterion based on volume ratios (formulated however in a different shape, with respect to what authors did):

Goix, N., Drougard, N., Brault, R., Chiapino, M.: One class splitting criteria for random forests. In: ACML, pp. 1–16 (2017)



**Strength And Weaknesses:**

Positive points
- The paper is well written and easy to read
- The topic is definitely interesting
- The structure of the manuscript is clear
- Experiments are extensive


Negative points
- Significance of the proposed method.
My main concern is about significance of the proposed method. It seems that authors start from ideas presented in PID and PIDforest, for anomaly detection, and optimized such framework and created an effective pipeline by adding several carefully chosen ingredients. The resulting framework is definitely well performing, as shown in the experiments, but I’m wondering how large is its methodological contribution from the Anomaly Detection perspective (but of course this is my personal opinion). As for the theoretical contribution, I think it is not so relevant, being reduced to the three lines of page 5 (by the way, the proof of the proposition, found in Appendix, is not clear to me, especially the sentence which concludes the proof)

- Conclusions from the experiments.
Experiments are very extensive, based on 20 tabular datasets and involving many different competitors. One comment which applies to the whole analysis. The tables show bold values, which are meant to highlight the best results; however these do not derive from a rigorous statistical analysis, but simply by the rule “Metrics with standard error overlapped with the best number are bolded” (from the paper, caption table 2). Without a rigorous statistical evaluation conclusive observations can not be derived. I suggest authors to use the Friedman test followed by a post-hoc Nemenyi test, and to show results via critical diagram – see for example

Demšar, Janez. "Statistical comparisons of classifiers over multiple data sets." The Journal of Machine learning research 7 (2006): 1-30.


**Summary Of The Paper:**

The paper presents an approach for tabular anomaly detection which is based on Partial Identification (PID) and on Generalized Additive Models (GAM) and extensions. The method works also in semisupervised settings and compares well with alternatives, as shown in a quite extensive experimental evaluation.

**Summary Of The Review:**

Paper which proposes a carefully tailored pipeline for anomaly detection of tabular data. Some doubts on its significance. Large experimental parts which should be completed with a rigorous statistical analysis.

UPDATE AFTER THE REBUTTAL.
I carefully read the responses, the clarifications, and the additional material, and I thank the authors for the significant efforts made in clarifying my doubts. Even if I consider that the paper has potential, I’ll maintain my score unchanged, since I still have few doubts, especially on the significance and on the analysis of the results (even if I acknowledge the efforts made by the authors in improving these aspects in the rebuttal)

---

> ### Author Response · Authors · 2022-11-19
> **Thank you for your review (1/2)**
>
> > Significance of the proposed method. My main concern is about significance of the proposed method. It seems that authors start from ideas presented in PID and PIDforest, for anomaly detection, and optimized such framework and created an effective pipeline by adding several carefully chosen ingredients. The resulting framework is definitely well performing, as shown in the experiments, but I’m wondering how large is its methodological contribution from the Anomaly Detection perspective (but of course this is my personal opinion).
>
> Thank you for being honest here. Indeed, the key ingredients of this method have been developed, but without the proposed key changes the method would not be as effective. For example, one of the key ingredients is the smoothing count – without increasing to a large number like 100, the model will in fact perform much worse.
>
> While combining PID with NodeGAM, we introduce multiple methodological innovations, such as estimating and normalizing a sparsity metric as the anomaly scores, integrating a regularization for an inductive bias appropriate for AD, and using deep representation learning via fine-tuning with a differentiable AUC loss. Overall, our innovations lead to strong empirical results -- DIAD outperforms other alternatives significantly, both in unsupervised and semi-supervised settings.
>
> We believe this is an understudied area (interpretable AD) and that introducing the interpretability of GAM model and fine-tuning under the labeled anomalies in the AD setting with a superior performance has demonstrated a key benefit.
>
>
>
> > As for the theoretical contribution, I think it is not so relevant, being reduced to the three lines of page 5 (by the way, the proof of the proposition, found in Appendix, is not clear to me, especially the sentence which concludes the proof)
>
> Thank you. We fixed the last sentence of the theoretical proof. The high level of the idea is that since our DIAD maximizes the variance of the sparsity but splitting on noise features would not increase such objective, it will then prefer splitting on others to increase its objective. Please let us know if there’s anything unclear.
>
>
>
> > "Conclusions from the experiments. Experiments are very extensive, based on 20 tabular datasets and involving many different competitors. One comment which applies to the whole analysis. The tables show bold values, which are meant to highlight the best results; however these do not derive from a rigorous statistical analysis, but simply by the rule “Metrics with standard error overlapped with the best number are bolded” (from the paper, caption table 2). Without a rigorous statistical evaluation conclusive observations can not be derived. I suggest authors to use the Friedman test followed by a post-hoc Nemenyi test, and to show results via critical diagram – see for example
> Demšar, Janez. ""Statistical comparisons of classifiers over multiple data sets."" The Journal of Machine learning research 7 (2006): 1-30."
>
>
> We thank you for your suggestions. Due to the time constraint of the rebuttal and our lack of statistical knowledge, we can not finish this comparison in time and there does not seem to have an easy implementation of the test and graphs described. If there’s an easy python package that does this, please let us know.
>
> With all that said, as achieving statistically significant improvements may not be easy due to the large standard deviation in these datasets, we may not expect that we achieved significant improvements. Instead, we show the superiority of the proposed method across 20 datasets by the average rank and performance, and our method brings unprecedented interpretability and fine-tuning to the AD.
>
>
> > Section 6.3 of the results (part “Explaining anomalous data): this section is not completely convincing. Are you extracting something that other methods can not extract?
>
>
>
> Yes. There’s no such explanation available before in the AD literature. Previous methods can only do local explanation such as gradients, LIME, SHAP that can only show approximated per-feature importance on a single sample. Since these techniques are often used to explain models that utilize higher-order feature interactions such as DNN, the results are just an approximation. In contrast, since DIAD is designed to have only 1st or 2nd feature interactions, our method can present global meaningful feature graphs that allows human to understand its global behavior and gain insights (e.g. pre and post fine-tuning on labeled anomalies). And its per-feature importance is exact i.e. there’s no approximation. To the best of our knowledge, we’re the first one to propose these global explanations in the AD literature. Also, the explainability in semi-supervised learning is also understudied even beyond AD. To demonstrate its benefit, we further include more visualizations in the Appendix H.

---

> > ### Author Response · Authors · 2022-11-19
> > **Reply (2/2)**
> >
> >
> >
> > > Moreover, plots in Fig 4 are too small, it is very difficult to get colors, lines and in general the content
> >
> > This is an unfortunate limitation of conference papers that limit the page length. We’ll provide the bigger figure in the Appendix for clarity.
> >
> >
> >
> > > "References: in general there are many citations to arxiv papers, also for old papers. Please try to avoid this whenever it is possible. There are also some duplicated entries:"
> >
> > Thank you. We have fixed all of them.
> >
> >
> > > "For what concerns novelty: Estimation of the PID in section 5. Maybe authors can consider also the work by Goix and colleagues, which extends Isolation Forests by optimizing a criterion based on volume ratios (formulated however in a different shape, with respect to what authors did):
> > Goix, N., Drougard, N., Brault, R., Chiapino, M.: One class splitting criteria for random forests. In: ACML, pp. 1–16 (2017)"
> >
> > Thank you for pointing out this paper! It seems that the notion of this one-class RF is quite similar to the spirit of PID to split the feature space into “dense” and “sparse” regions. We believe these 2 criteria should be quite related to each other and should perform similarly. We believe our goal is not to provide some novel analysis as the above paper does, but instead propose a powerful method that allows explanations and fine-tuning in the AD setting.

---

### Author Response · Authors · 2022-11-19
**Summary of the revision**

Thank you! We have included more visualizations in Appendix H, more comprehensive noise injection experiments in Appendix I, and further compare with the Generalized Isolation Forest in Table 2 and 3. We respond to each comments below.

---

### Author Response · Authors · 2022-12-05
**Gentle reminder: discussions end at 12/12**

Dear all reviewers,

Please let us know if our response has addressed your concerns. We appreciate your time and efforts!

---

### Decision · Program_Chairs · 2023-01-20

**Decision:**

Reject

**Justification For Why Not Higher Score:**

Some changes made in combing two existing methods (PID and GAM) are marginal, so I suggest it not to be accepted in its current form.

**Justification For Why Not Lower Score:**

N/A

**Metareview: Summary, Strengths And Weaknesses:**

This paper presents a tabular anomaly detection method, referred to as DIAD, which tackles both interpretability and leveraging small amount of labeled examples. It bases its development on partial identification (PID) and generalized additive model (GAM). Certainly, some critical changes, instead of just combining existing methods, are proposed to make the method effective. However, most of reviewers feel that such changes made in combining PID and GAM are not substantial. In addition, the evaluations in the aspect of interpretable anomaly detection are weak. While the authors made efforts in responding to reviewers’ concerns, the main concern on the limited novelty kept them to stood by their earlier decisions. Therefore, the paper is not recommended for acceptance in its current form. I hope authors found the review comments informative and can improve their paper by addressing these carefully in future submissions.